# Myosin V functions as a vesicle tether at the plasma membrane to control neurotransmitter release in central synapses

Dario Maschi[1,2†], Michael W Gramlich[1,2†], Vitaly A Klyachko[1,2*]

[1]Department of Cell Biology and Physiology, Washington University, Missouri, United States; [2]Department of Biomedical Engineering, Washington University, Missouri, United States

**Abstract** Synaptic vesicle fusion occurs at specialized release sites at the active zone. How refilling of release sites with new vesicles is regulated in central synapses remains poorly understood. Using nanoscale-resolution detection of individual release events in rat hippocampal synapses we found that inhibition of myosin V, the predominant vesicle-associated motor, strongly reduced refilling of the release sites during repetitive stimulation. Single-vesicle tracking revealed that recycling vesicles continuously shuttle between a plasma membrane pool and an inner pool. Vesicle retention at the membrane pool was regulated by neural activity in a myosin V dependent manner. Ultrastructural measurements of vesicle occupancy at the plasma membrane together with analyses of single-vesicle trajectories during vesicle shuttling between the pools suggest that myosin V acts as a vesicle tether at the plasma membrane, rather than a motor transporting vesicles to the release sites, or directly regulating vesicle exocytosis.
DOI: https://doi.org/10.7554/eLife.39440.001

**\*For correspondence:**
klyachko@wustl.edu

[†]These authors contributed equally to this work

**Competing interests:** The authors declare that no competing interests exist.

## Introduction

Quantal vesicle release at the synaptic active zone (AZ) represents a unitary event of information transmission at synapses. Recent nanoscale resolution studies revealed that vesicle release is organized in multiple discreet release sites, which are distributed throughout the AZ and spatially coincide with the clusters of presynaptic docking factors (*Maschi and Klyachko, 2017*; *Tang et al., 2016*). While the spatial organization and molecular architecture of release sites are beginning to emerge, the mechanisms that dynamically regulate the release site properties and reuse remain poorly understood.

Following a fusion event, refilling of a release site with a new vesicle is believed to represent a critical rate-limiting step in the release site reuse capacity thereby governing the ability of synapses to sustain release during repetitive activity (*Neher, 2010*). How this process is organized and regulated at the AZ is largely unknown due to the difficulty of directly visualizing the refilling process at the AZ, whosedimensions are typically at or below the diffraction limited resolution of conventional microscopy.

Presynaptic terminals are rich in actin filaments (*Cingolani and Goda, 2008*) and recent studies suggested that refilling of release sites in cerebellar synapses is actin-dependent (*Miki et al., 2016*). The major unanswered question is what role actin cytoskeleton and actin-based vesicle transport play in release site refilling: an active, structural, or both? An active, actin-dependent role occurs via molecular motors, for which only myosin V has been identified as a presynaptic vesicle-associated protein in central neurons (*Prekeris and Terrian, 1997*; *Takamori et al., 2006*; *Watanabe et al.,*

*2005*). Major synaptic defects were observed in cortical neurons of myosin V dominant-negative mutant mice *flailer* (*Yoshii et al., 2013*) and due to myosin V knockdown (*Correia et al., 2008*). Myosin V null mice exhibit severe seizures and human mutations in myosin V gene cause severe nervous system dysfunction known as Griscelli syndrome (*Hammer and Wagner, 2013*; *Kneussel and Wagner, 2013*; *Pastural et al., 1997*). These findings point to a major role of myosin V in synaptic function; yet whether or how myosin V regulates neurotransmitter release remains largely unexplored. Single-vesicle tracking experiments showed that a majority of recycling vesicles undergo large-scale motion within synaptic boutons (*Forte et al., 2017*; *Gramlich and Klyachko, 2017*; *Kamin et al., 2010*; *Lee et al., 2012*; *Park et al., 2012*; *Peng et al., 2012*; *Westphal et al., 2008*) including a substantial component of directed actin-dependent motion (*Forte et al., 2017*; *Gramlich and Klyachko, 2017*; *Peng et al., 2012*). Myosin V may thus regulate release by supporting vesicle transport to the release sites. In addition to being a processive motor, myosin V is also known to function as a tether and to interact with the SNARE proteins in a $Ca^{2+}$-dependent manner to promote the SNARE complex formation (*Krementsov et al., 2004*; *Ohyama et al., 2001*; *Prekeris and Terrian, 1997*; *Watanabe et al., 2005*), leading several studies to suggest a role for myosin V in vesicle docking and exocytosis (*Desnos et al., 2007*; *Eichler et al., 2006*; *Porat-Shliom et al., 2013*; *Rudolf et al., 2011*). Whether in central synapses myosin V plays a role in the vesicle transport to refill release sites, or at the later stages of the refilling process by tethering the vesicle to the release site machinery, or in the vesicle exocytosis itself is poorly understood.

Here, we sought to address these questions by employing a nanoscale detection of individual vesicle release events at the AZ in the hippocampal boutons, together with single-vesicle tracking to visualize release site refilling and reuse. Our results uncover a major role for myosin V in release site refilling, but not the exocytosis process itself. Surprisingly, rather than a unidirectional vesicle flow towards the release sites, we observed a dynamic vesicle shuttling between a plasma membrane pool and an inner pool, which is regulated by neuronal activity and requires myosin V as a vesicle tether rather than a transporting motor. These results, supported by ultrastructural analyses, suggest a major role for myosin V in regulating neurotransmitter release by controlling vesicle retention at the release sites rather than vesicle transport to the release sites or the exocytosis process itself.

## Results

### Inhibition of myosin V reduces release site re-use in hippocampal synapses

To understand the role of myosin V in presynaptic release mechanisms, we employed a nanoscale imaging modality to examine the effects of myosin V inhibition on spatiotemporal features of individual release events in hippocampal boutons. Our imaging approach takes advantage of a pH-sensitive indicator pHluorin targeted to the vesicle lumen via vGlut1 (vGlut1-pHluorin) (*Balaji and Ryan, 2007*; *Leitz and Kavalali, 2011*; *Voglmaier et al., 2006*) to permit detection of single vesicle release events with a 27 nm precision (*Maschi and Klyachko, 2017*). vGlut1-pHluorin was expressed in cultures of excitatory hippocampal neurons using a lentiviral infection at DIV3 and imaging was performed at DIV 16 – 19 at 37°C. Robust detection of individual release events evoked by 1 AP stimulation at 1 Hz was achieved within individual synapses at 40 ms/frame rate throughout the observation time period of 120 s (*Figure 1A*). Hierarchical clustering algorithms were used to define individual release sites within each bouton using a cluster diameter of 50 nm (*Maschi and Klyachko, 2017*).

To avoid potential developmental effects of interfering with myosin V function, we used the most acute approach to inhibit myosin V in developed neurons using two different and highly specific agents that arrest the myosin V ATP/ADP cycle (MyoVin-1 (Myo1) and Pentabromopseudilin (PBP)) (*Fedorov et al., 2009*; *Gramlich and Klyachko, 2017*; *Islam et al., 2010*). Using this approach, we examined the effects of myosin V inhibition on vesicle release in basal conditions and during repetitive stimulation to distinguish two hypothesized functions of myosin V in synaptic transmission:

- Myosin V may regulate refilling of the release sites with a new vesicle. In this case, myosin V inhibition would not strongly affect the basal release probability from a resting state when release sites are presumed to be largely occupied. In contrast, we expect to observe a significant effect of myosin V inhibition on vesicle release probability during repetitive activity. This

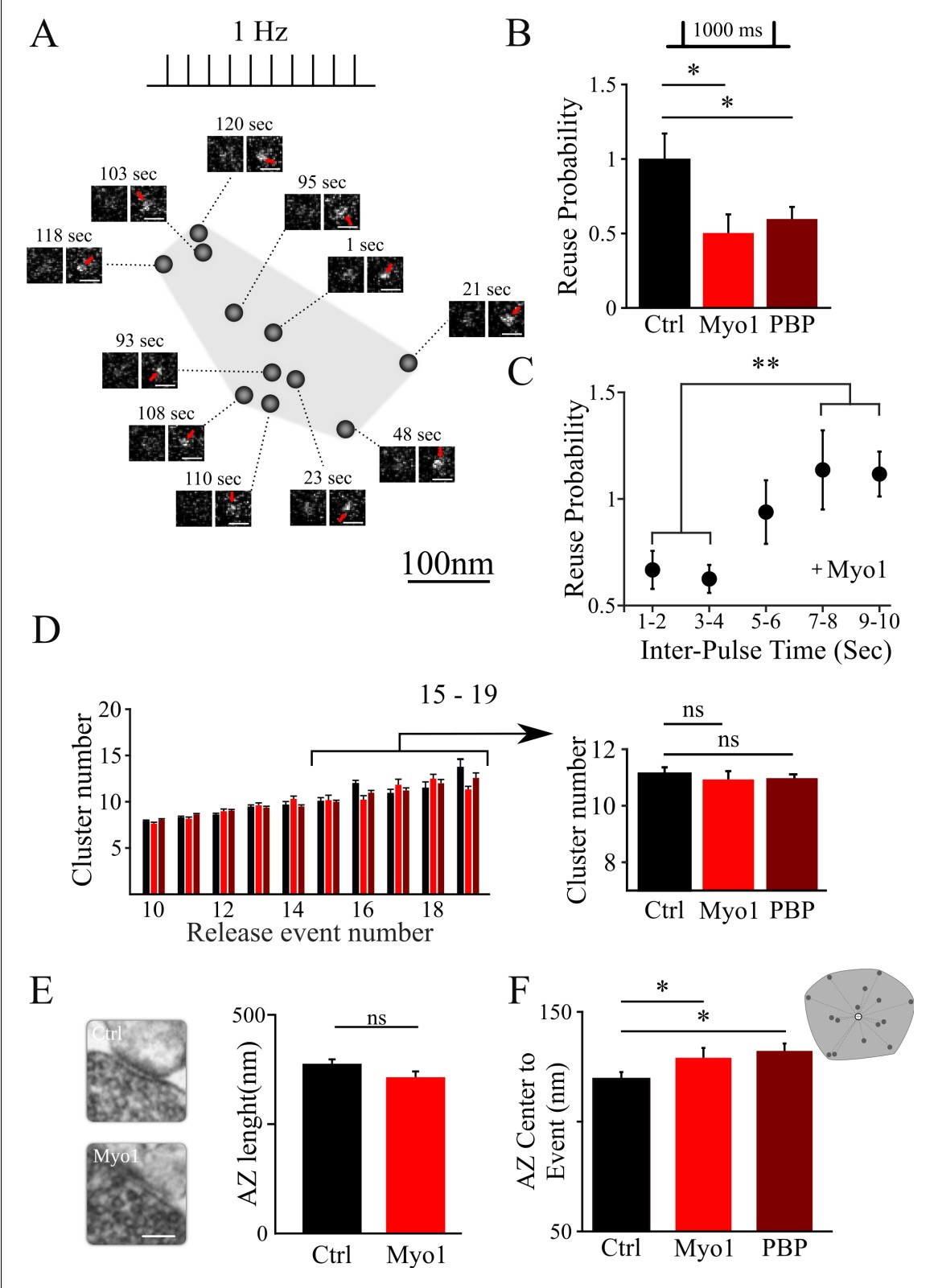

**Figure 1.** Inhibition of myosin V impairs the reuse of release sites in hippocampal synapses. (**A**) Sample spatial distribution of release events within a single hippocampal bouton evoked by 1 Hz stimulation. Scale bar: 1 µm. (**B**) Effect of myosin V inhibition with Myo1 or PBP on release site reuse was evaluated using a paired-pulse protocol as a probability that the same site is reused for two sequential stimuli 1000 ms apart, normalized to the same measurement in control condition. (**C**) Effect of myosin V inhibition with Myo1 on the probability that the same site is reused for two stimuli at different
*Figure 1 continued on next page*

*Figure 1 continued*

inter-stimulus intervals (in the range of 1 – 10 s) normalized to the same measurement in control conditions and binned for two sequential intervals. (D) Effect of myosin V inhibition on the average number of clusters/release sites detected in individual boutons plotted as a function of the number of release events observed (left). Pooled average number of clusters for boutons with 15 to 19 detected events in different conditions indicated (right). (E) Large-Area Scanning Electron Microscopy (LaSEM) of hippocampal boutons in culture showing that no significant changes in the AZ size were detected as a result of myosin V inhibition with Myo1. Examples of control and Myo1 treatment are shown (left). Scale bar: 200 nm. (F) Effect of myosin V inhibition on the distribution of distances of release events to AZ center evoked at 1 Hz. **=P < 0.01, *=P < 0.05, two-sample t-test; ns – not significant.

DOI: https://doi.org/10.7554/eLife.39440.002

The following figure supplement is available for figure 1:

**Figure supplement 1.** DMSO alone has no effect on release site reuse.

DOI: https://doi.org/10.7554/eLife.39440.003

effect should be more pronounced as stimulation frequency increases when demand for the new vesicles is higher.

- Myosin V may function downstream of release site refilling, as a mediator of the exocytosis process itself. In this case, the requirement for myosin V should be apparent independently of the stimulation frequency. Thus, inhibition of myosin V is expected to reduce the vesicle release probability from a resting state as well as during repetitive activity, and the magnitude of the effect should be similar for a wide range of stimulus frequencies.

To discern these models, we first examined effect of myosin V inhibition on basal release probability, which we estimated based on a number of single release events evoked by 100 APs at 1 Hz. Inhibition of myosin V with either MyoVin-1 or PBP had no or only a small effect on basal release probability (*Table 1*), arguing against the major role for myosin V in the exocytosis process itself.

Next, we examined how inhibition of myosin V affects the reuse of individual release sites using a paired-pulse protocol at 1000 ms (*Maschi and Klyachko, 2017*). In this analysis, for each stimulus pair, we identified a subset of boutons in which release events were detected for both stimuli in the pair; we then determined the probability that the two release events occur at the same release site. This protocol measures vesicle refilling of the release site on short timescales. We found that both myosin V inhibitors, MyoVin-1 or PBP, markedly reduced the reuse probability of release sites (*Figure 1B*; *Table 1*), while DMSO alone had no effect (*Figure 1—figure supplement 1*, *Table 1*). Furthermore, we observed that this effect of myosin V inhibition dissipated with increase in time between release events, and was no longer apparent when release events were separated by more than ~5 s (*Figure 1C*, *Table 1*). These findings suggest that myosin V regulates rapid refilling of release sites. However, on longer timescales, once the release sites are refilled, the release process itself appears not to depend on myosin V.

The change in release site reuse probability could reflect the role of myosin V in refilling the sites with a new vesicle, but this effect could also arise if the number of release sites is altered by myosin V inhibition. We thus examined whether inhibition of myosin V affects the number and basic structural organization of release sites. The number of clusters/release sites in individual boutons was not affected by myosin V inhibition (*Figure 1D*, *Table 1*). Ultrastructural analysis using a Large-Area Scanning Electron Microscopy (LaSEM) further indicated that the AZ size was also not altered by myosin V inhibition (*Figure 1E*, *Table 1*). Interestingly, we noted a small, but significant effect of myosin V inhibition on increasing the average distance from release events to the AZ center (*Figure 1F*, *Table 1*). One possible interpretation of this result is that peripheral release sites are engaged more frequently when myosin V is inhibited (see below for additional analyses).

Thus far these results favor the first model that myosin V has an important role in the refilling of release sites rather than the exocytosis itself.

## Myosin V controls release site refilling during high-frequency stimulation

If myosin V plays a role in refilling of the release sites, inhibition of myosin V should have a larger effect on release magnitude at higher stimulation frequencies when demand for vesicles is increased. To determine if this is the case, we measured the magnitude of release at individual boutons (as given by the total vGlut1-pHluorin signal) evoked by high-frequency trains. Both MyoVin-1 and PBP reduced the magnitude of synaptic release evoked by 20 stimuli at 50 Hz to half compared with the

**Table 1.** Table of all data values and statistical analyses.

Data table columns are formatted as (i) corresponding figure location; (ii) conditions being statistically compared and separated by '/'; (iii) number of samples (synapses, dishes, cultures) used for each test; (iv) mean values and errors for each condition separated by '/' and corresponding to conditions in column (i); (v) statistical test used for comparison; (vi) P-value resulting from the statistical comparison.

| Figure number | Conditions | NSyn | NDishes | Ncultures | mean ± sem | Stastical test | P val |
|---|---|---|---|---|---|---|---|
| Basal Pr (related to 1) | Ctrl/Myo-1 | 367/259 | 15/15 | 4/3 | 0.064 ± 0.001/0.056 ± 0.002 | Two-sample t-test | 0.004 |
| | Ctrl/PBP | 367/839 | 15/11 | 4/3 | 0.064 ± 0.001/0.067 ± 0.001 | Two-sample t-test | 0.15 |
| 1(B) | Ctrl/Myo-1 | 367/259 | 15/15 | 4/3 | 1.000 ± 0.006/0.503 ± 0.004 | Two-sample t-test | 0.02 |
| | Ctrl/PBP | 367/839 | 15/11 | 4/3 | 1.000 ± 0.006/0.5971 ± 0.003 | Two-sample t-test | 0.02 |
| 1(C) | 1–4/7–10 | 367/259 | 15/15 | 4/3 | 0.63 ± 0.06/1.036 ± 0.1028 | Two-sample t-test | 0.003 |
| 1(D) | Ctrl/Myo-1 | 367/259 | 15/15 | 4/3 | 11.1 ± 0.2/10.9 ± 0.3 | Two-sample t-test | 0.58 |
| | Ctrl/PBP | 367/839 | 15/11 | 4/3 | 11.1 ± 0.2/11.0 ± 0.1 | Two-sample t-test | 0.56 |
| 1(E) | Ctrl/Myo-1 | 137/81 | 3/3 | 3/3 | 378 ± 10/364 ± 13 | Two-sample t-test | 0.38 |
| 1(F) | Ctrl/Myo-1 | 367/259 | 15/15 | 4/3 | 120 ± 3/129 ± 4 | Two-sample t-test | 0.04 |
| | Ctrl/PBP | 367/839 | 15/11 | 4/3 | 120 ± 3/132 ± 3 | Two-sample t-test | 0.01 |
| 2(B) | Ctrl/Myo-1 | - | 9/10 | 3/3 | 0.50 ± 0.06 | Two-sample t-test | <0.001 |
| | Ctrl/PBP | - | 9/7 | 3/3 | 0.64 ± 0.04 | Two-sample t-test | <0.001 |
| Slope (related to 2B) | Ctrl/Myo-1 | - | 9/10 | 3/3 | 0.52 ± 0.05 | Two-sample t-test | <0.001 |
| | Ctrl/PBP | - | 9/7 | 3/3 | 0.62 ± 0.04 | Two-sample t-test | <0.001 |
| 2(C) | Ctrl/Myo-1 | - | 36/34 | 11/11 | y = 0.75269–0.17806 x (Linear Fit) | Linear Fit | <0.001 |
| 2(F) | 1 Hz/10 Hz, Ctrl | 367/254 | 15/15 | 4/6 | 120 ± 3/126 ± 2 | Two-sample t-test | 0.04 |
| | 1 Hz/10 Hz, Myo-1 | 259/862 | 15/10 | 3/3 | 129 ± 4/151 ± 6 | Two-sample t-test | 0.004 |
| | 1 Hz/10 Hz, PBP | 839/988 | 11/11 | 3/3 | 132 ± 3/145 ± 4 | Two-sample t-test | 0.01 |
| 3(C) | Ctrl/Myo-1, KCl | 86/67 | 3/3 | 3/3 | 43.19 ± 0.02/59.32 ± 0.03 | Two-sample KS-test | <0.001 |
| 3(D) | Ctrl/Myo-1, KCl | 86/67 | 3/3 | 3/3 | 37 ± 3/13 ± 3 | Two-sample t-test | <0.001 |
| 5(C) Baseline | Ctrl(-)/Ctrl(+) | 29/122 | 27/64 | 5/12 | 0.048 ± 0.017/0.05 ± 0.0012 | Two-sample KS-test | 0.42 |
| | Ctrl(-)/Myo-1(+) | 29/69 | 27/40 | 5/8 | 0.048 ± 0.017/0.047 ± 0.01 | Two-sample KS-test | 0.42 |
| | Ctrl(-)/PBP(+) | 29/21 | 27/47 | 5/8 | 0.048 ± 0.017/0.048 ± 0.008 | Two-sample KS-test | 0.42 |
| | Ctrl(-)/EGTA(+) | 29/51 | 27/30 | 5/6 | 0.048 ± 0.017/0.03 ± 0.006 | Two-sample KS-test | 0.88 |
| | Ctrl(-)/DMSO(+) | 29/44 | 27/34 | 5/5 | 0.048 ± 0.017/0.046 ± 0.01 | Two-sample KS-test | 0.43 |
| | Ctrl(+)/Myo-1(+) | 122/69 | 64/40 | 12/8 | 0.05 ± 0.0012/0.047 ± 0.01 | Two-sample KS-test | 0.43 |
| | Ctrl(+)/PBP(+) | 122/21 | 64/47 | 12/8 | 0.05 ± 0.0012/0.048 ± 0.008 | Two-sample KS-test | 0.43 |
| | Ctrl(+)/EGTA(+) | 122/51 | 64/30 | 12/6 | 0.05 ± 0.0012/0.03 ± 0.006 | Two-sample KS-test | 0.88 |
| | Ctrl(+)/DMSO(+) | 122/44 | 64/34 | 12/5 | 0.05 ± 0.0012/0.04608 ± 0.01053 | Two-sample KS-test | 0.13 |
| | Myo-1(+)/PBP(+) | 69/21 | 40/47 | 8/8 | 0.047 ± 0.01/0.04831 ± 0.00749 | Two-sample KS-test | 0.88 |
| | Myo-1(+)/EGTA(+) | 69/51 | 40/30 | 8/6 | 0.047 ± 0.01/0.03201±0.00639 | Two-sample KS-test | 0.43 |
| | PBP(+)/EGTA(+) | 21/51 | 47/30 | 8/6 | 0.048 ± 0.008/0.03 ± 0.006 | Two-sample KS-test | 0.43 |

*Table 1 continued on next page*

*Table 1 continued*

| Figure number | Conditions | NSyn | NDishes | Ncultures | mean ± sem | Stastical test | P val |
|---|---|---|---|---|---|---|---|
| 5(C) 20 Hz stimulation | Ctrl(-)/Ctrl(+) | 29/122 | 27/64 | 5/12 | 0.026 ± 0.002/0.013 ± 0.0003 | Two-sample KS-test | <0.001 |
| | Ctrl(-)/Myo-1(+) | 29/69 | 27/40 | 5/8 | 0.026 ± 0.002/0.029 ± 0.003 | Two-sample KS-test | 0.16 |
| | Ctrl(-)/PBP(+) | 29/21 | 27/47 | 5/8 | 0.026 ± 0.002/0.03 ± 0.002 | Two-sample KS-test | 0.63 |
| | Ctrl(-)/EGTA(+) | 29/51 | 27/30 | 5/6 | 0.026 ± 0.002/0.013 ± 0.002 | Two-sample KS-test | <0.001 |
| | Ctrl(+)/Myo-1(+) | 122/69 | 64/40 | 12/8 | 0.013 ± 0.0003/0.02907 ± 0.00264 | Two-sample KS-test | 0.02 |
| | Ctrl(+)/PBP(+) | 122/21 | 64/47 | 12/8 | 0.013 ± 0.0003/0.03 ± 0.002 | Two-sample KS-test | 0.02 |
| | Ctrl(+)/EGTA(+) | 122/51 | 64/30 | 12/6 | 0.013 ± 0.0003/0.013 ± 0.002 | Two-sample KS-test | 0.63 |
| | Ctrl(+)/DMSO(+) | 122/44 | 64/34 | 12/5 | 0.013 ± 0.0003/0.012 ± 0.0006 | Two-sample KS-test | 0.66 |
| | Myo-1(+)/PBP(+) | 69/21 | 40/47 | 8/8 | 0.029 ± 0.003/0.03 ± 0.002 | Two-sample KS-test | 0.16 |
| | Myo-1(+)/EGTA(+) | 69/51 | 40/30 | 8/6 | 0.029 ± 0.003/0.013 ± 0.002 | Two-sample KS-test | 0.02 |
| | PBP(+)/EGTA(+) | 21/51 | 47/30 | 8/6 | 0.03 ± 0.002/0.013 ± 0.002 | Two-sample KS-test | 0.007 |
| 5(B) | MC-Model (-) | 100 | 1000 | | 8.50E-02 | | |
| | MC-Model (+) | 100 | 1000 | | 3.00E-02 | | |
| 5(E) Baseline | Ctrl(-)/Ctrl(+) | 60/343 | 27/64 | 5/12 | 0.02 ± 0.003/0.02 ± 0.01 | Two-sample KS-test | 0.14 |
| | Ctrl(-)/Myo-1(+) | 60/174 | 27/49 | 5/8 | 0.02 ± 0.003/0.02 ± 0.002 | Two-sample KS-test | 0.59 |
| | Ctrl(-)/PBP(+) | 60/55 | 27/34 | 5/8 | 0.02 ± 0.003/0.02 ± 0.005 | Two-sample KS-test | 0.14 |
| | Ctrl(-)/EGTA(+) | 60/63 | 27/30 | 5/6 | 0.02 ± 0.003/0.016 ± 0.005 | Two-sample KS-test | 0.14 |
| | Ctrl(+)/Myo-1(+) | 343/174 | 64/40 | 12/8 | 0.02 ± 0.01/0.02 ± 0.002 | Two-sample KS-test | 1 |
| | Ctrl(+)/PBP(+) | 343/55 | 64/34 | 12/8 | 0.02 ± 0.01/0.02 ± 0.005 | Two-sample KS-test | 0.59 |
| | Ctrl(+)/EGTA(+) | 343/63 | 64/30 | 12/6 | 0.02 ± 0.01/0.016 ± 0.005 | Two-sample KS-test | 0.31 |
| | Myo-1(+)/PBP(+) | 174/55 | 40/34 | 8/8 | 0.02 ± 0.002/0.02 ± 0.005 | Two-sample KS-test | 0.89 |
| | Myo-1(+)/EGTA(+) | 174/63 | 40/30 | 8/6 | 0.02 ± 0.002/0.016 ± 0.005 | Two-sample KS-test | 0.14 |
| 5(E) 20 Hz stimulation | Ctrl(-)/Ctrl(+) | 60/343 | 27/64 | 5/12 | 0.025 ± 0.001/0.02 ± 0.0006 | Two-sample KS-test | 0.59 |
| | Ctrl(-)/Myo-1(+) | 60/174 | 27/49 | 5/8 | 0.025 ± 0.001/0.025 ± 0.0005 | Two-sample KS-test | 1 |
| | Ctrl(-)/PBP(+) | 60/55 | 27/34 | 5/8 | 0.025 ± 0.001/0.024 ± 0.003 | Two-sample KS-test | 0.31 |
| | Ctrl(-)/EGTA(+) | 60/63 | 27/30 | 5/6 | 0.025 ± 0.001/0.015 ± 0.001 | Two-sample KS-test | 0.001 |
| | Ctrl(+)/Myo-1(+) | 343/174 | 64/40 | 12/8 | 0.02 ± 0.0006/0.025 ± 0.0005 | Two-sample KS-test | 0.89 |
| | Ctrl(+)/PBP(+) | 343/55 | 64/34 | 12/8 | 0.02 ± 0.0006/0.024 ± 0.003 | Two-sample KS-test | 0.31 |
| | Ctrl(+)/EGTA(+) | 343/63 | 64/30 | 12/6 | 0.02 ± 0.0006/0.015 ± 0.001 | Two-sample KS-test | 0.001 |
| | Myo-1(+)/PBP(+) | 174/55 | 40/34 | 8/8 | 0.025 ± 0.0005/0.024 ± 0.003 | Two-sample KS-test | 0.14 |
| | Myo-1(+)/EGTA(+) | 174/63 | 40/30 | 8/6 | 0.025 ± 0.0005/0.015 ± 0.001 | Two-sample KS-test | 0.001 |
| Displacement (related to 6B) | Ctrl(-) Last 5/First 5 s | 12 | 27 | 5 | 1.17 ± 0.12 | Two-sample t-test | 0.001 |
| | Ctrl(+) Last 5/First 5 s | 35 | 64 | 12 | 1.08 ± 0.05 | Two-sample t-test | 0.02 |
| | Myo-1(+) Last 5/First 5 s | 24 | 40 | 8 | 1.18 ± 0.12 | Two-sample t-test | <0.001 |
| 6(D) Change in Velocity | Ctrl(-)/Ctrl(-) | 12 | 27 | 5 | 1.54 ± 0.13 | Two-sample t-test | <0.001 |
| | Ctrl(+)/Ctrl(+) | 35 | 64 | 12 | 1.2 ± 0.016 | Two-sample t-test | 0.022 |
| | Myo-1(+)/Myo-1(+) | 24 | 40 | 8 | 1.31 ± 0.02 | Two-sample t-test | <0.001 |
| | PBP(+)/PBP(+) | 17 | 47 | 8 | 1.27 ± 0.03 | Two-sample t-test | 0.002 |
| | EGTA(+)/EGTA(+) | 20 | 30 | 6 | 1.29 ± 0.02 | Two-sample t-test | 0.002 |

*Table 1 continued on next page*

*Table 1 continued*

| Figure number | Conditions | NSyn | NDishes | Ncultures | mean ± sem | Stastical test | P val |
|---|---|---|---|---|---|---|---|
| 6(E) Change in Angle | Ctrl(-)/Ctrl(-) | 12 | 27 | 5 | 1.02 ± 0.04 | Two-sample KS-test | 1 |
| | Ctrl(+)/Ctrl(+) | 35 | 64 | 12 | 0.95 ± 0.04 | Two-sample KS-test | 1 |
| | Myo-1(+)/Myo-1(+) | 24 | 40 | 8 | 0.88 ± 0.03 | Two-sample KS-test | 0.65 |
| | PBP(+)/PBP(+) | 17 | 47 | 8 | 1.03 ± 0.04 | Two-sample KS-test | 1 |
| | EGTA(+)/EGTA(+) | 20 | 30 | 6 | 1.06 ± 0.04 | Two-sample KS-test | 1 |
| *Figure 1— figure supplement 1(A)* | Ctrl/DMSO | - | 15/8 | 4/3 | 1.00 ± 0.17/0.97 ± 0.16 | Chi-squared test | 0.89 |
| *Figure 2— figure supplement 1(A)* | Ctrl/DMSO | - | 7/9 | 3/3 | 1.0 ± 0.1/0.86 ± 0.09 | Two-sample t-test | 0.43 |
| *Figure 2— figure supplement 1(B)* | Ctrl/Myo-1 | - | 9/10 | 3/3 | 2.74 ± 0.5/3.42 ± 0.98 | Two-sample t-test | 0.56 |
| | Ctrl/PBP | - | 9/7 | 3/3 | 2.74 ± 0.5/1.47 ± 1.18 | Two-sample t-test | 0.3 |
| *Figure 2— figure supplement 1(C)* | Ctrl/Myo-1, KCl | 86/67 | 3/3 | 3/3 | 38 ± 3/14±3 | Two-sample t-test | 0.40 |
| *Figure 2— figure supplement 1(E)* | Ctrl/Myo-1 | - | 9/10 | 3/3 | 2.05 ± 0.16/2.06 ± 0.11 | Two-sample t-test | 0.93 |
| | Ctrl/PBP | - | 9/7 | 3/3 | 2.05 ± 0.16/2.39 ± 0.07 | Two-sample t-test | 0.09 |
| *Figure 2— figure supplement 1(F)* | Ctrl/Myo-1 | - | 9/10 | 3/3 | 3.6 ± 0.1/3.42 ± 0.06 | Two-sample t-test | 0.2 |
| | Ctrl/PBP | - | 9/7 | 3/3 | 3.6 ± 0.1/3.51 ± 0.07 | Two-sample t-test | 0.57 |
| *Figure 2— figure supplement 1(G)* | Ctrl/Myo-1 | - | 9/10 | 3/3 | y = 0.50915–0.0016552 x (Linear Fit) | Linear Fit | 0.57 |
| | Ctrl/Myo-1; 1 st vs 2–5 | - | 9/10 | 3/3 | 0.52 ± 0.06/0.50 ± 0.02 | Two-sample t-test | 0.81 |
| *Figure 2— figure supplement 1(H)* | Ctrl/PBP | - | 9/7 | 3/3 | y = 0.61391 + 0.00052562 x (Linear Fit) | Linear Fit | 0.93 |
| | Ctrl/PBP; 1 st vs 2–5 | - | 9/7 | 3/3 | 0.61 ± 0.04/0.62 ± 0.02 | Two-sample t-test | 0.85 |
| *Figure 3— figure supplement 1(A)* | Ctrl/Myo-1, KCl | 86/67 | 3/3 | 3/3 | 43.23 ± 0.02/41.15 ± 0.02 | Two-sample KS-test | <0.001 |
| *Figure 3— figure supplement 1(B)* | Ctrl/Myo-1 | 137/81 | 3/3 | 3/3 | 34 ± 2/33 ± 3 | Two-sample t-test | 0.84 |
| *Figure 4— figure supplement 1(H)* | Ctrl Disappear/Re-appear | 16/16 | 64/64 | 12/12 | 1.14 ± 0.105/0.88 ± 0.062 | Two-sample t-test | 0.04 |

DOI: https://doi.org/10.7554/eLife.39440.004

control (*Figure 2A,B*, *Table 1*), while DMSO alone had no effect (*Figure 2—figure supplement 1A*, *Table 1*). Inhibition of myosin V also slowed the kinetics of release to a similar degree (as evident by the change in the slope of the vGlut1-pHluorin signal increase during stimulation, *Table 1*), which is consistent with slower refilling of the release sites when myosin V is inhibited. We further examined how release magnitude is affected in a wide range of stimulus frequencies from 0.1 Hz to 50 Hz. We found that inhibition of myosin V had a larger effect on the magnitude of release at higher stimulus frequencies while it did not have any measurable effect on release at a very low 0.1 Hz stimulation frequency (*Figure 2C*, *Table 1*). The observation that release amplitude is normal at very low stimulus frequencies supports the notion that the exocytosis process itself is not strongly affected by myosin V inhibition.

In addition to these temporal requirements for myosin V in supporting release, our result above suggested that spatial distribution of release may be altered by myosin V inhibition. We thus further examined this function of myosin V during high-frequency stimulation. Inhibition of myosin V significantly exacerbated the use of peripheral release sites during high frequency stimulation (*Figure 2D, E, F*, *Table 1*) suggesting its role in controlling not only temporal but also the spatial distribution of release site reuse. One plausible explanation for the spatial effects of myosin V inhibition is its involvement in endocytosis independently of its role in refilling of release sites. If this is the case, inhibition of myosin V could cause an increase in the distance of peripheral release sites to the center because of an activity-dependent accumulation of vesicle components on the plasma membrane and an increase in the AZ size. Ultrastructural EM analysis showed no significant changes in the AZ size by myosin V inhibition both at baseline (*Figure 1E*) and during KCl-induced depolarization

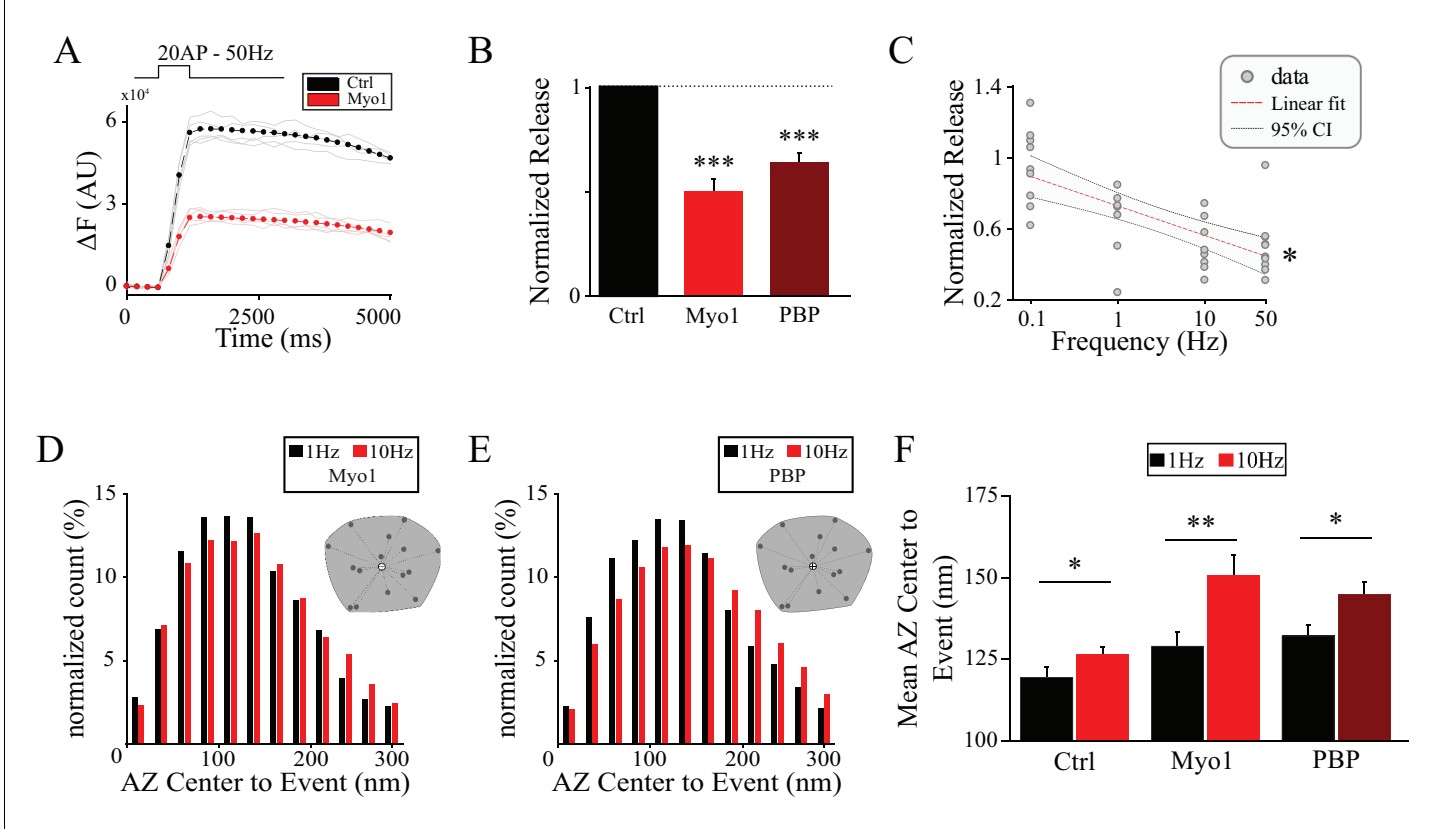

**Figure 2.** Myosin V controls release site refilling during high-frequency stimulation. (A) Examples of vGlut1-pHluorin responses at single hippocampal boutons to 20 stimuli trains at 50 Hz for Ctrl (black) and Myo1 (red). (B) Ratio of vGlut1-pHluorin responses in (A) at individual boutons for Myo1 or PBP normalized to the control. (C) Effect of myosin inhibition with Myo1 on the amplitude of vGlut1-pHluorin responses to 20 stimuli trains at individual boutons as a function of the train frequency, averaged across all boutons in the movie and normalized to a control at a corresponding frequency. (D–E) Effect of myosin inhibition with Myo1 (D) or PBP (E) on the distribution of distances of release events to AZ center evoked at 1 Hz vs 10 Hz. (F) Analysis of data in (D,E) showing a mean distance of release events to the AZ center per bouton at 1 Hz or 10 Hz for Ctrl, Myo1 and PBP, respectively. ***=P < 0.001, **=P < 0.01, *=P < 0.05, two-sample t-test; ns – not significant.

DOI: https://doi.org/10.7554/eLife.39440.005

The following figure supplement is available for figure 2:

**Figure supplement 1.** Controls for vesicle release measurements during high-frequency stimulation.

DOI: https://doi.org/10.7554/eLife.39440.006

(*Figure 2—figure supplement 1B*) arguing against major effects of myosin V inhibition on endocytosis. To confirm this observation, we further examined if inhibition of myosin V affects endocytosis in live and active synapses. The decay of vGlut1-pHluorin signal following a stimulus train is determined by endocytosis and subsequent vesicle reacidification. Changes in the decay of the vGlut1-pHluorin signal in our measurements can thus be interpreted to reflect predominantly changes in endocytosis (*Atluri and Ryan, 2006*). We found that inhibition of myosin V had no measurable effect on the decay of the vGlut1-pHluorin signal following 50 Hz trains (*Figure 2—figure supplement 1C-E*). Moreover, if inhibition of myosin V blocks endocytosis, application of sequential stimulus trains would be expected to cause surface accumulation of VGlut1-pHluorin and a corresponding increase in the bouton fluorescence. We compared the amplitudes of VGlut1-pHluorin signal in five consecutive 50 Hz trains separated by 20 s each, and did not observe any measurable changes from one train to the next under control conditions or in the presence of myosin V inhibitors (*Figure 2—figure supplement 1F-H*). Combined, these results indicate that the shift in spatial localization of release events upon myosin V inhibition is not caused by an increase in the AZ size and is unlikely to be mediated by altered endocytosis.

Together these observations support the first model in which myosin V plays a major role in regulating the refilling of the release sites rather than the exocytosis or endocytosis processes themselves. Our results further suggest a role for myosin V in regulating spatial distribution of release, with a shift towards more peripheral release sites when myosin V is inhibited.

## Inhibition of myosin V causes a vesicle docking defect during sustained activity

If myosin V indeed plays a role in release site refilling, we predicted that inhibition of myosin V should reduce the number of docked vesicles at the AZ during sustained activity. We tested this idea using a scanning electron microscopy (LaSEM) of hippocampal cultures treated or not with Myo-Vin-1 for 20 min and stimulated via a KCl-induced depolarization for 10 min (with a 10 min delay after beginning of MyoVin-1 treatment), and fixed immediately thereafter (*Figure 3A*). Experiments were accompanied by the corresponding control measurements without KCl depolarization. We quantified the effect of myosin V inhibition on vesicle localization by dividing the AZ into 0.5 nm segments and calculating the distance from each segment to the closest vesicle (*Figure 3B*). The distribution of these minimal distance values provides a measure of the vesicle location relative to the AZ, and has an additional advantage of being independent of the total number of vesicles detected in each bouton. We observed that vesicles were localized significantly farther away from the AZ when myosin V was inhibited during sustained depolarization with KCl (*Figure 3A,C*, *Table 1*), but this effect was not apparent in cultures not treated with KCl (*Figure 3—figure supplement 1A*, *Table 1*).

To further quantity changes in vesicle localization caused by inhibition of myosin V, we examined the ratio of 'docked' vesicles and more loosely 'tethered' vesicles: docked vesicles were defined as vesicles adjacent to the AZ, with the vesicle center within 30 nm from the AZ (which is equivalent to vesicle membrane being less than ~5 nm from the AZ assuming a vesicle diameter of 50 nm, and within an error of measurement of our LaSEM that has a pixel size of 5 nm), while 'tethered' vesicles included all vesicles with a center within 100 nm from the AZ. Only the subset of vesicles identified above as being the closest to the AZ were used for this analysis to be independent of the total number of vesicles detected in each bouton. We found that the ratio of 'docked' to 'tethered' vesicles at the AZ was markedly reduced by myosin V inhibition during sustained depolarization with KCl (*Figure 3D*, *Table 1*), but not in the absence of KCl (*Figure 3—figure supplement 1B*, *Table 1*).

Mechanistically, the role of myosin V in controlling vesicle localization and release site refilling can involve its function as a vesicle transporting motor to the release sites and/or as a tether of vesicles at the release sites or both. Below we employed a single vesicle tracking approach to discern these possibilities.

## Tracking individual synaptic vesicles during recycling supports a model of continuous vesicle shuttling between two vesicle pools

To better understand the release site refilling process, we used our established approach for tracking individual synaptic vesicles during recycling and translocation to the AZ (*Forte et al., 2017*; *Gramlich and Klyachko, 2017*; *Peng et al., 2012*). Individual vesicles were labeled with a lipophilic FM-like dye SGC5 via compensatory endocytosis using a pair of stimuli at 100 ms. Due to very low basal release probability (~0.06, *Table 1*) of hippocampal synapses at 37°C, this protocol labels either none or at most a single vesicle in vast majority of boutons (*Forte et al., 2017*; *Peng et al., 2012*); a small subset of boutons with two or more detected vesicles were excluded from further analyses. This imaging approach permits us to track individual vesicles with ~20 nm precision within hippocampal boutons (*Figure 4A*) (*Forte et al., 2017*; *Peng et al., 2012*).

We previously found that the majority of vesicles undergo transitions between epochs of directed, diffusive and stalled motion during recycling (*Forte et al., 2017*). We and others also previously reported that vesicle mobility decreased with time, which has been interpreted to represent vesicle settling within one of the functional vesicle pools (*Kamin et al., 2010*), a membrane-bound readily-releasable pool (RRP) or an inner recycling or reserved pool. Here, we observed that a large subset of these settled vesicles (>50% during 2 min observation) became more mobile and then disappeared during our observation window (*Figure 4A,B*). The same (or possibly another) vesicle was also often observed (re-) appearing in the same bouton with some delay (*Figure 4D,E*). These

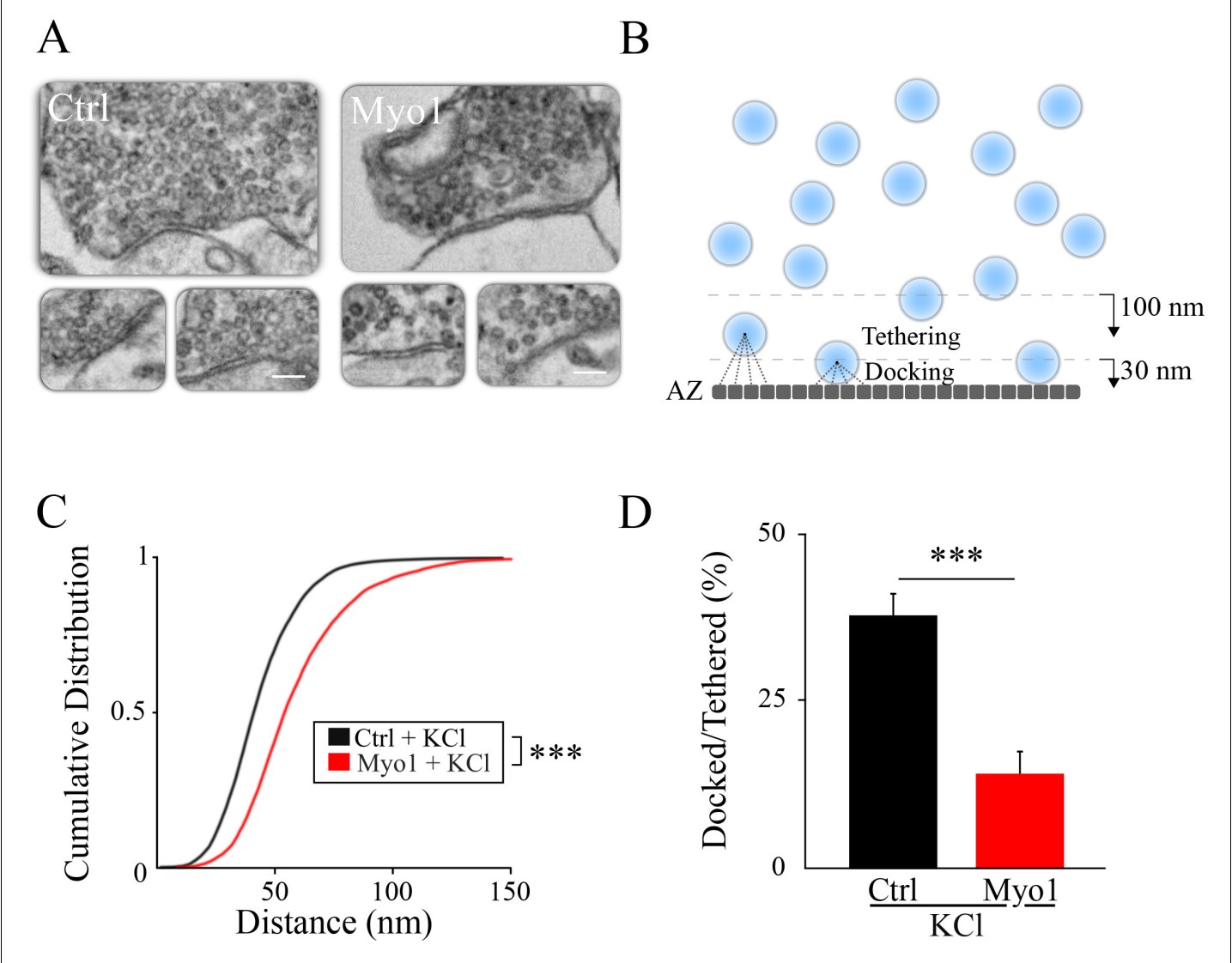

**Figure 3.** Inhibition of myosin V causes a vesicle docking defect during sustained activity. (**A**) LaSEM of individual hippocampal boutons in cultures depolarized by KCl application (55 mM) for 10 min in the presence or absence of Myo1 (20 min), immediately followed by fixation. (**B–D**) Membrane opposite to the PSD was divided into 0.5 nm sections (**B**) and the distance from each section to the closest vesicle was determined and plotted as a cumulative histogram (**C**). This subset of closest vesicles was subsequently used to estimate the relation between docked and tethered vesicle populations (**D**). We considered vesicle as 'docked' when the distance from the AZ section to the vesicle center was under 30 nm and 'tethered' when the distance was under 100 nm. Scale bar: 200 nm. ***=P < 0.001; **=P < 0.01; two-sample KS-test (**C**) or two-sample t-test (**D**). ns = not significant.

DOI: https://doi.org/10.7554/eLife.39440.007

The following figure supplement is available for figure 3:

**Figure supplement 1.** Ultrastructural analysis of vesicle docking under basal conditions in the absence of KCl.

DOI: https://doi.org/10.7554/eLife.39440.008

appearance and disappearance events were often observed multiple times in the same bouton (*Figure 4D,E*) and both occurred at constant rates, with the rate of disappearance ~2 – 3 fold higher than the rate of appearance (*Figure 5B-E*, *Table 1*; the fraction of disappearing vesicles per second (δ-rate): 0.046 ± 0.016 fraction/sec; fraction of appearing vesicles per second (α-rate): 0.022 ± 0.010 fraction/sec; see Materials and Methods for quantification details). We note that because disappearance is the only criteria for the vesicle inclusion in our initial analysis, the initial conditions for calculating the disappearance and appearance rates are not equivalent. As a result, the fractional curves in (*Figure 5B,D*) representing these two processes are visually different, although the actual rates

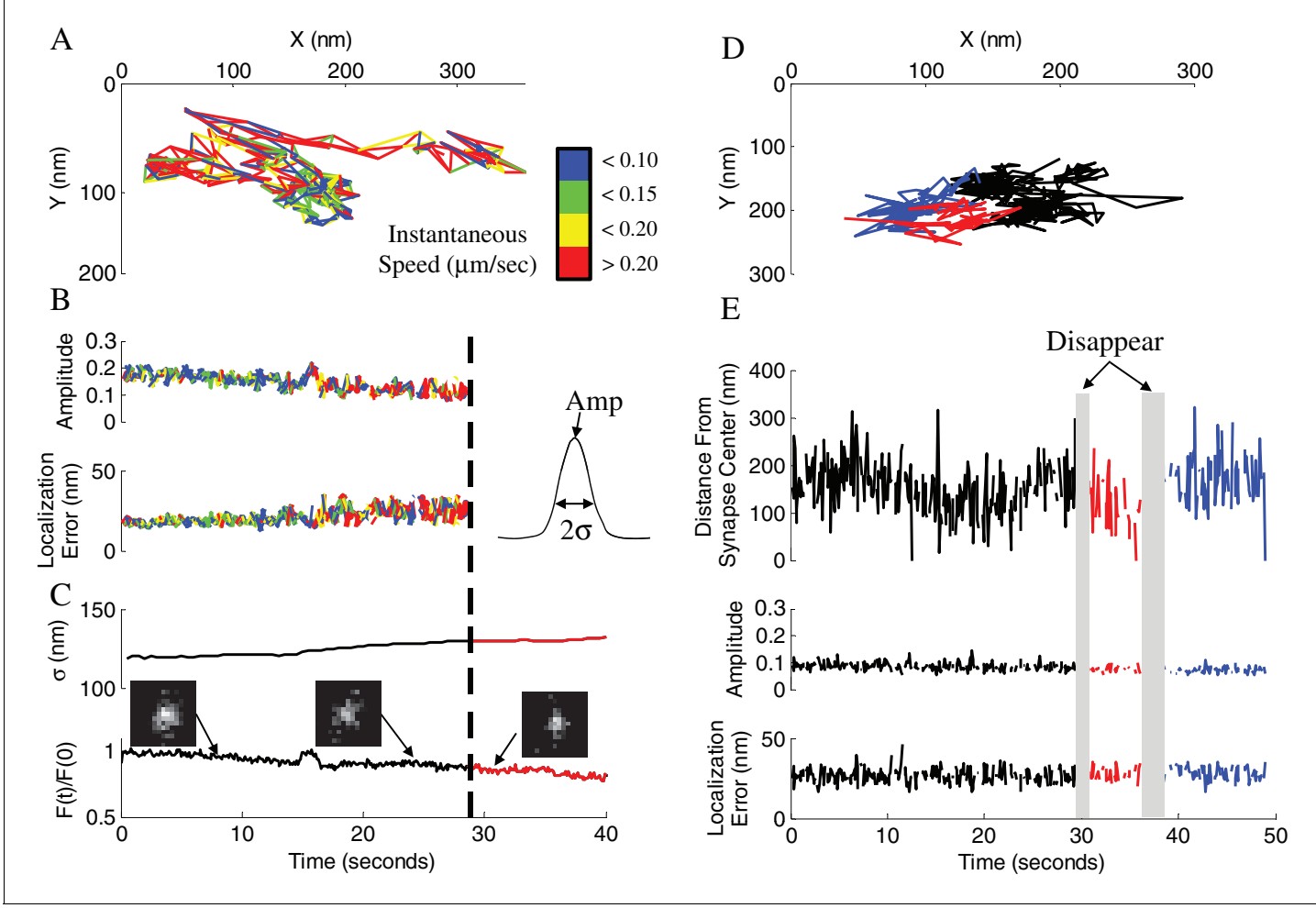

**Figure 4.** Vesicle disappearance/reappearance events. (A) Sample track of a vesicle that disappeared during observation. Track is color-coded by instantaneous vesicle speed. (B) Parameters of vesicle image from (A) before and after vesicle disappearance indicated by a dotted line. Amplitude of the vesicle image (PSF) normalized to image bit-depth ($2^{14}$-1) (*Top*), and localization error of vesicle position as a function of time (*Bottom*). (C) Half-width of a Gaussian fit to the whole-synapse image ($\sigma$, *Top*) and whole-synapse integrated intensity (*Bottom*) before (Black) and after (Red) vesicle detection was lost, for the same vesicle as in A. Raw images (insets) of vesicle image at different time points show that vesicle is still present in the synapse after detection is lost. (D) Sample track of a vesicle that undergoes multiple disappearance/reappearance events. Track is color-coded with initial track shown in black, the first reappearance in red, the second reappearance in blue. (E) Parameters of vesicle track from (D). Vesicle position over time plotted as a distance from center of the bouton (*Top*). Amplitude of the vesicle image (*Middle*) and localization error of vesicle position (*Bottom*) as a function of time. Periods of vesicle disappearance are highlighted in grey. The color coding is the same as in (D).

DOI: https://doi.org/10.7554/eLife.39440.009

The following figure supplement is available for figure 4:

**Figure supplement 1.** Controls for vesicle tracking analyses.

DOI: https://doi.org/10.7554/eLife.39440.010

are constant in both cases. Specifically, only vesicles that were detected at T = 0 were counted in disappearance analysis *and* only a single disappearance event was counted per bouton. To be able to distinguish the disappearance and reappearance events, the appearance analysis was limited to the same subset of synapses in which vesicle disappearance was observed first; but this analysis placed *no* limitation on the number of vesicle appearance events counted in the same bouton. As a result of these initial conditions, the number of labeled vesicles that could disappear is maximal at T = 0 and is continuously drawn down with time leading to a visually apparent reduction in the disappearance rate (*Figure 5B*). This effect is caused by un-labeled vesicles replacing the labeled ones and also disappearing at the same rate but without being counted. This reduces the apparent

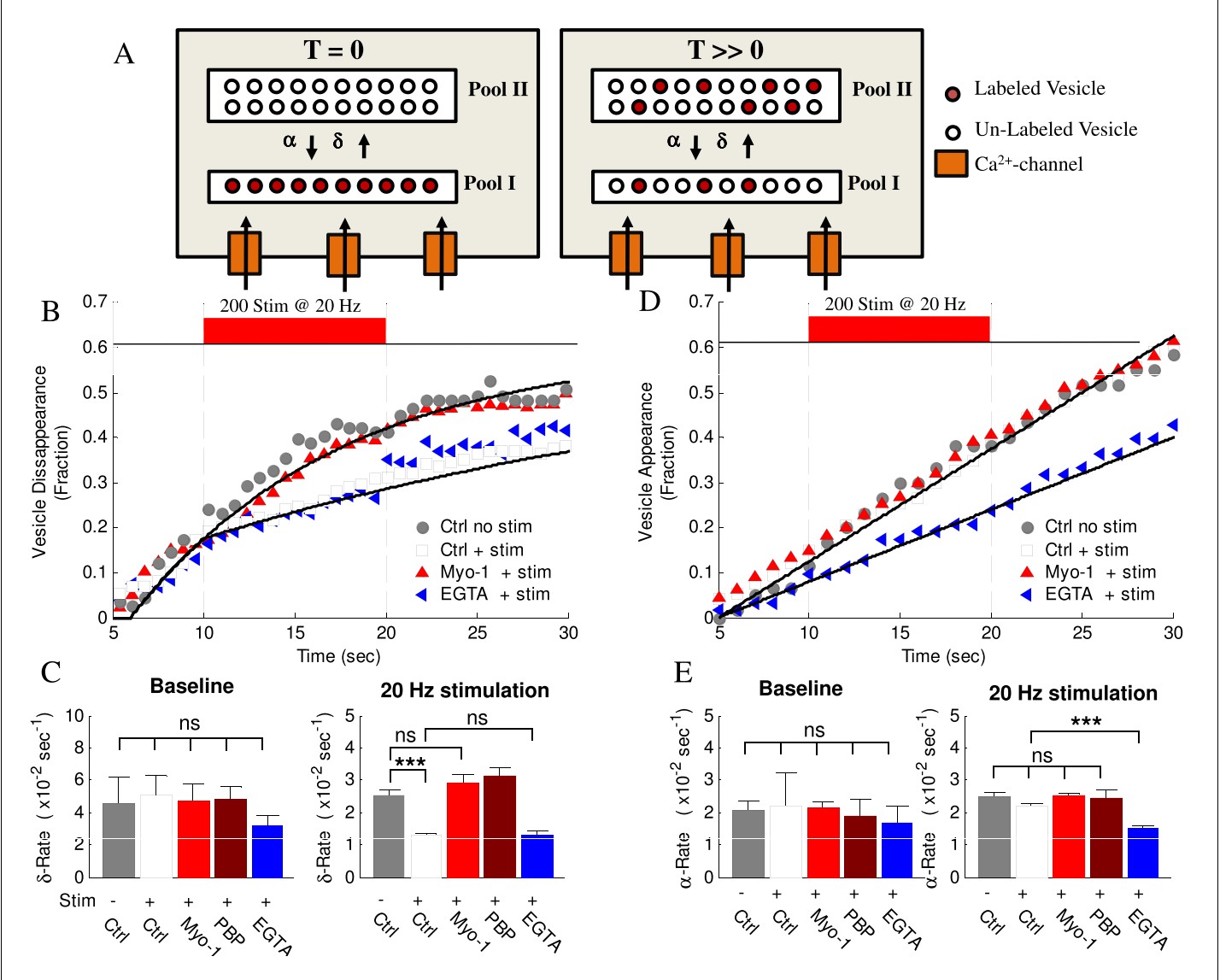

**Figure 5.** Activity- and myosin V- dependent vesicle shuttling between a membrane and an inner pools. (A) Model hypothesis of vesicle exchange between two pools resulting in observed disappearance and re-appearance. At T = 0 all labeled vesicles are assumed to be in Pool I; vesicles undergo transition toward Pool II at rate (δ) and re-appear with rate (α). With time (T >> 0) the experimentally observed δ–rate is lower because un-labeled vesicles replace the labeled ones and disappear at the same rate but without being counted. (B) Vesicle disappearance as a function of time plotted as a fraction of total vesicles observed. Ctrl-No Stim represents vesicle disappearance rate in the absence of activity at any time during observation; other data show effects of stimulation (+Stim, 20 Hz applied at 10 – 20 s period) without (white) or with (red) myosin-V inhibition (Myo-1), or EGTA-AM (blue). Computational model is shown as solid lines. Note that the X-axis starts at T = 5 s because all tracks were required to be observed for at least 5 s to be included in analysis. (C) Mean disappearance rate (δ) from exponential-recovery fits to data in (B) at baseline (5–10 s) and during stimulation (10 – 20 s). Error-bars are mean-residual of fits to data. (D) Same as (B) but for vesicle appearances in different conditions indicated plotted as a fraction of total vesicles observed. Linear fits to data show constant rates (solid lines). (E) Same as (C) for the mean appearance rate (α) from linear fits to data in (D). Error-bars are mean-residual of linear fits to data. ***=P < 0.001; two-sample KS-test. ns = not significant.
DOI: https://doi.org/10.7554/eLife.39440.011

disappearance rate, while the actual rate remains constant. In contrast, these limits on initial conditions do not affect the appearance rate because the number of labeled vesicles that can disappear is constantly replenished and appearance events are counted every time even if they appear multiple times in the same bouton. Consequently, the appearance rate remains linear.

Because vesicle disappearance and reappearance is often observed multiple times in the same bouton, we hypothesized that this process represents motion of the same vesicle in and out of focus,

since very few boutons have more than one labeled vesicle present, and there is no any additional source of the dye other than the vesicle that was initially labeled. These vesicle disappearance/appearance observations may thus be mechanistically explained by a model hypothesis of vesicle transitions between two distinct pools (*Figure 5A*), one of which is in the focal plane, and the other one is outside of the focal plane. If the total number of vesicles exchanged between the two pools is constant in our model, the α-rate and δ-rate infer the fraction of total vesicles in each pool, then the 2 – 3 fold differences in the α- and δ-rates can be accounted for by assuming that the Pool II is 2 – 3 times larger than Pool I. Experiments below are designed to provide evidence to support this 2-pool model and will examine how vesicle mobility between (and retention at) these pools are regulated by activity and myosin V. We will then provide evidence that the smaller Pool I is a RRP-like membrane pool localized close to the source of calcium influx into the synapse, while the larger Pool II is farther away from the calcium influx and thus represents an inner pool.

While we hypothesize that vesicle disappearance can be explained by loss of detection due to large-scale displacement away from the focal plane within the synaptic bouton, vesicle disappearance can also be caused by either exocytosis or a vesicle leaving the synapse via axonal transport in a process known as an inter-synaptic vesicle exchange (*Gramlich and Klyachko, 2017*). To distinguish these possibilities we quantified changes in the integrated intensity of the entire synapse in which vesicle disappearance occurred (*Figure 4C*). The total synapse intensity arises predominantly from the intensity of the labeled vesicle and a nonspecific background; it is thus expected to undergo a stepwise decrease upon vesicle fusion and loss of the dye. We found that the integrated synapse intensity remained unchanged at the time point of vesicle disappearance (*Figure 4C* and *Figure 4—figure supplement 1D*) for the vast majority of disappearing vesicles (~97%). Further, the few cases when integrated intensities did decrease matched the gradual intensity decrease profile observed for inter-synaptic vesicle exchange events (*Figure 4—figure supplement 1A-D*). These results suggest that the vast majority of disappearance events are not caused by exocytosis, but rather by loss of detection due to vesicle moving out of focus.

The shape of the vesicle image is represented by the point-spread function (PSF), which becomes increasingly broader as vesicle goes out of focus (*Figure 4C*) eventually causing loss of detection when a preset threshold is reached. This notion is evident in the corresponding increase in the PSF σ (half-width) at the time points of vesicle disappearance and a decrease in the PSF σ at time points of subsequent vesicle reappearance (*Figure 4—figure supplement 1H*). These changes were not caused by differential bleaching of vesicle signal vs background (*Figure 4—figure supplement 1I*). We were also able to correlate the observed range of vesicle PSF σ values (~120 – 150 nm) with the range of detectable vesicle motion along the Z-axis of ~100 nm (*Figure 4—figure supplement 1G*), suggesting that the two vesicle pools are at least 100 nm apart from each other. To further clarify the causes of vesicle disappearance we re-analyzed the trajectories of disappearing vesicles under less strict detection criteria by allowing detection of the more out of focus vesicles with broader images (*Figure 4—figure supplement 1E,F*). We found that with less stringent detection criteria, the vast majority of disappearing vesicles could be tracked beyond the initial disappearance point, supporting the notion that loss of tracking occurred due to vesicle motion out focus rather than fusion. Finally, we observed that the vesicle velocity and displacement both increased significantly in the last 2 s before disappearance, consistent with vesicle acceleration immediately prior to disappearance (*Figure 6D*, *Table 1*). These results provide further evidence that vesicle disappearance was caused by loss of detection due to vesicle motion away from the focal plane rather than vesicle fusion.

Using less strict detection criteria and thus the extended ability to track vesicle trajectories beyond the initial disappearance point, we also confirmed our integrated intensity analysis above showing that a small fraction (<3%) of the disappearance events was caused by vesicle leaving the synapse and entering the axon (*Figure 4—figure supplement 1A-D*), which is a process we have previously reported (*Gramlich and Klyachko, 2017*). Thus vesicle disappearance is not caused by travel out of the synapse in >97% of the cases. We note that generally, such inter-synaptic vesicle exchange events occur more frequently than the 3%, but the majority of these events do not result in loss of tracking and are automatically excluded from all our current analyses because their displacement is beyond the spatial limits we set for the synapse size (displacement >0.8 μm from the synapse center).

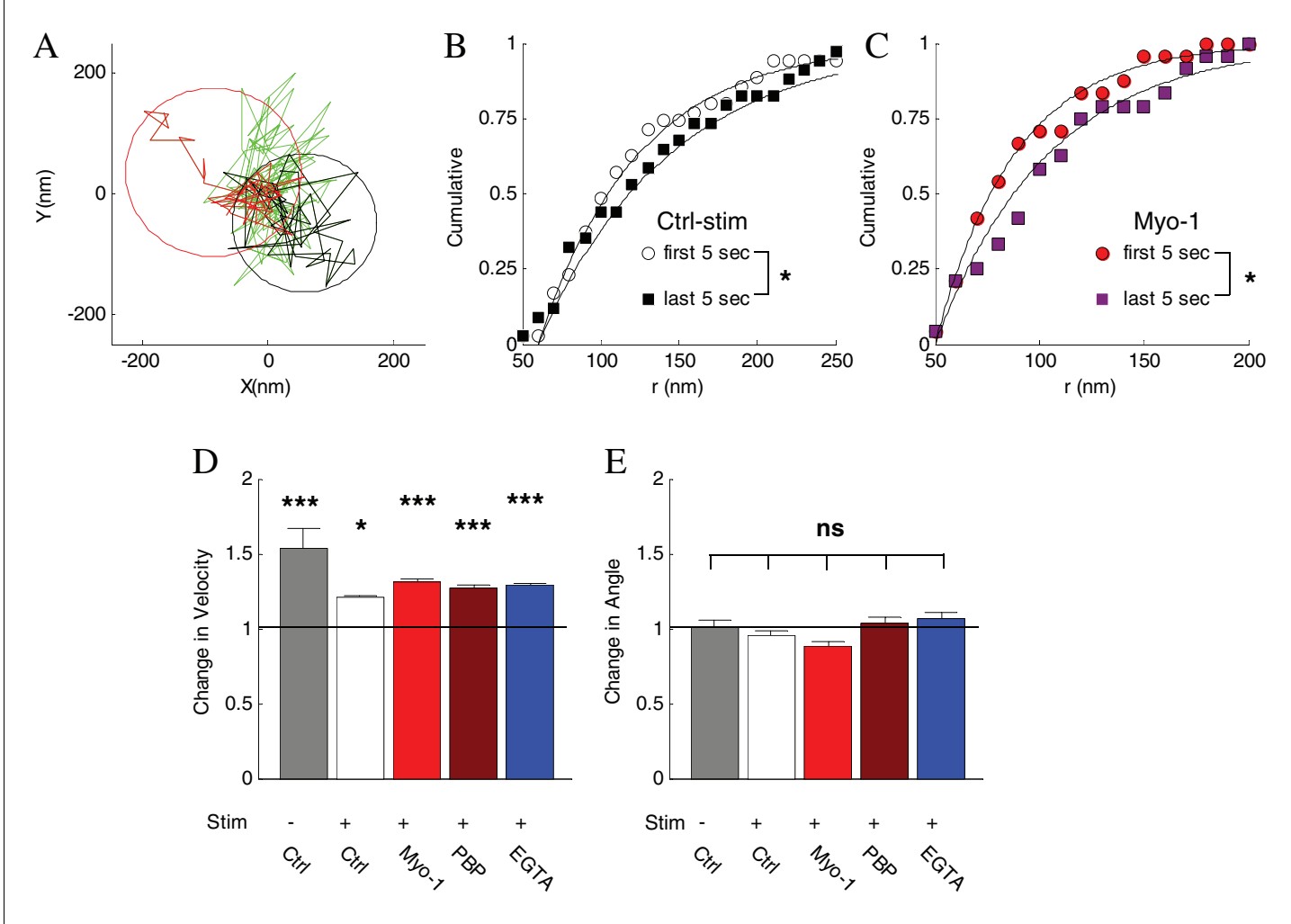

**Figure 6.** Inhibtion of myosin V does not affect vesicle mobility before disappearance. (A) Spatial analysis of vesicle displacement based on the radius of a circle encompassing 95% of vesicle trajectory for the first 5 s (50 frames, black circle) and for the last 5 s (red circle) before disappearance. (B) Cumulative distributions for the spatial analysis in (A) in control with stimulation conditions compared for all disappearing vesicles and fit to exponential recovery function (solid line). (C) Same as (B) in the presence of Myo-1 and stimulation. (D) The ratio of instantaneous speed during the last 2 s and the first 2 s of observation. Error-bars are determined from residuals of cumulative fits. (E) The ratio of angular displacement during the last 2 s and the first 2 s of observation. Error-bars are determined from SEM. ***=P < 0.001; *=P < 0.05; two-sample t-test (D) or two-sample KS-test (E).
DOI: https://doi.org/10.7554/eLife.39440.012

Taken together these results indicate that in the vast majority of cases, the observed vesicle disappearance is not caused by exocytosis or travel out of the synapse, but rather by vesicle acceleration and motion to another pool located out of focus; subsequent vesicle re-appearance represses vesicle transition back.

## Vesicle shuttling between the pools is activity- and myosin V dependent

We previously found that a component of vesicle mobility during recycling is regulated in an activity-dependent manner; however which step in the vesicle cycle this component corresponds to is unknown. Here we found that the rate of vesicle disappearance was reduced 2 – 3-fold during high-frequency stimulation (*Figure 5B,C*, *Table 1*). This activity-evoked change in the rate had a very fast onset (<0.5 s) but persisted long after the end of stimulation (*Figure 5B*), consistent with the rapid rise of presynaptic calcium at the beginning of stimulus trains and a slow calcium clearance following the cessation of stimulation (*Neher and Sakaba, 2008*). Importantly, we observed that inhibition of

myosin V with either Myo1 or PBP eliminated the effect of activity on vesicle disappearance during stimulation, without affecting the baseline disappearance rate before stimulation (*Figure 5B,C*, *Table 1*), while DMSO alone had no effect (*Table 1*). In contrast, the rate of subsequent vesicle appearance in the same set of boutons was not strongly affected by either high-frequency stimulation or myosin V inhibition (*Figure 5D,E*, *Table 1*). These results suggest that myosin V mediates the activity-dependence of vesicle disappearance, but not vesicle re-appearance.

To better understand the vesicle disappearance, we used Monte Carlo simulations to model this process (*Figure 5A*, and solid lines in 5B, *Table 1*). Vesicle disappearance rate was calculated using a computational model that simulates individual vesicle disappearances from a pool of vesicles (see cartoon in *Figure 5A* and Appendix for model details). A single constant disappearance probability captures the baseline vesicle disappearance rate ($P_{\delta, baseline}$= 8.5×10$^{-2}$ fraction/sec). This result suggests that vesicles are randomly exchanged in and out of focus without preference for residence time (i.e. newly arrived vesicles are just as likely to leave as established vesicles). The model shows that the experimentally observed reduction in disappearance rate upon stimulation can be modeled with an instantaneous 3-fold reduction in disappearance probability (from $P_{\delta, base-line}$ = 8.5×10$^{-2}$ fraction/sec, to $P_{\delta, stim}$ = 3.0×10$^{-2}$ fraction/sec) and the effect of myosin V inhibition is captured by an increase in the disappearance probability during stimulation back to the baseline rate ($P_{\delta}$=8.5×10$^{-2}$ fraction/sec). These results support the above observations that the vesicle disappearance is rapidly modulated by activity and that this modulation is myosin V dependent.

## Vesicle shuttling represents transitions between a membrane pool and an internal pool

How are vesicle disappearance and reappearance related to vesicle location inside the synaptic bouton and specifically the AZ? We hypothesized that the smaller Pool I, which is rapidly regulated by activity/calcium influx, is an RRP-like membrane vesicle pool localized close to the source of calcium influx into the synapse, while the larger Pool II represents an inner pool, further away from the calcium sources. The 2 – 3 fold difference in size between the Pools I and II in our model is also consistent with a smaller membrane-bound RRP and larger internal recycling pool described previously (*Denker and Rizzoli, 2010*).

To test this model prediction and relate the location of the two pools to the source of calcium influx, we used a slow calcium chelator EGTA. EGTA is too slow to buffer rapid calcium rise in the close proximity to voltage-gated calcium channels at the plasma membrane, but is effective in buffering subsequent slow calcium elevation in the interior of the bouton due to diffusion. Indeed, it has been shown that EGTA does not affect vesicle release, but effectively blocks facilitation and augmentation, the two forms of short-term synaptic enhancement that dependent on residual calcium elevation during repetitive activity (*Deng and Klyachko, 2011*; *Regehr, 2011*). We thus pre-incubated neurons with a cell-permeable EGTA-AM for 20 min. Calcium buffering with EGTA had no effect on the activity-dependent change in the disappearance rate during high-frequency stimulation (*Figure 5B,C*, *Table 1*) suggesting that vesicle disappearance occurs from a pool located in a close proximity to the source of calcium influx, that is the AZ. In contrast, the appearance rate in the same set of boutons was significantly reduced by EGTA indicating efficiency of EGTA treatment (*Figure 5D,E*, *Table 1*). These results suggest that in contrast to vesicle disappearance, the appearance represents transition from a pool located farther away from the source of calcium influx. We note that although we did not detect a measurable effect of activity on the appearance rate, it is possible that the calcium-dependence of this process is already close to saturation even in the basal conditions, but is unmasked by lowering the basal calcium levels by EGTA.

The finding that EGTA strongly affects only the vesicle appearance rate (and thus vesicles in Pool II) also provides a simple way to estimate the ratio of vesicles that start in each pool in beginning of our observations. Since disappearance is the only criteria for the vesicle inclusion in our initial analysis *and* due to the arbitrary relationship between the focal plane and the AZ at any given synapse, there is no *a priori* information that would allow us to make the initial assignment of the vesicles to Pool I or Pool II. Our measurements of the disappearance rate thus reflect a mixture of vesicles that are in Pool I and Pool II at the beginning of observation. To perform this analysis, we assumed that only the vesicle appearance rate is truly affected by EGTA (based on our observation that disappearance rate for EGTA and Ctrl during stimulation are the same), while the small effect of EGTA on the disappearance rate (vesicles in Pool I) is caused by contribution from Pool II vesicles. Within these

assumptions, the model (Appendix) and EGTA results suggest that among all vesicles assumed to be in Pool I in the beginning of observations, 75 ± 11% actually start in Pool I, while 25 ± 11% are physically located in Pool II. This result is consistent with the Pool II being larger, and having a slower exit rate, because the probability that a given labeled single vesicle in our experiment will leave the Pool II is smaller than if it would be in Pool I and thus less likely to be included in our disappearance analysis.

Together these results support the model that vesicle shuttling occurs between the smaller membrane and larger internal pools, with vesicle transitions away from (but not towards) the plasma membrane pool being both activity- and myosin V dependent. In contrast with widely assumed model of unidirectional vesicle flow during recycling towards the AZ, our results suggest an ongoing vesicle shuttling and exchange between an inner pool and a membrane pool.

## The role of myosin V in vesicle tethering at the membrane pool vs vesicle transport

Our results suggest that vesicle disappearance rate (translocation away from the membrane pool) is strongly reduced by activity in a manner that requires myosin V. Myosin V, a vesicle-associated protein, may function in this process as a vesicle-transporting motor and/or as a tethering/docking factor that retains vesicles in a membrane pool in an activity-dependent manner. To distinguish these possibilities, we examined whether inhibition of myosin V alters vesicle mobility immediately prior to disappearance.

First, we employed a spatial analysis of vesicle displacement we developed previously based on calculating the radius of a circle encompassing 95% of the vesicle track (*Figure 6A*) (*Forte et al., 2017*; *Peng et al., 2012*). We compared the spatial extent of vesicle motion over the first 5 s of observations and the last 5 s before disappearance. Vesicle displacement increased significantly just before disappearance (*Figure 6B*), and the magnitude of the increase was similar in all conditions (*Table 1*). Importantly, this increase in displacement was not affected by inhibition of myosin V (*Figure 6C*, *Table 1*).

We corroborated this result by examining changes in the instantaneous vesicle velocity during a 2 s period immediately before disappearance compared to the first 2 s of observation because analysis of instantaneous vesicle velocity permits higher time resolution than analysis of spatial displacement (which provides a more global assessment of vesicle mobility). Consistent with the above results, the vesicle velocity increased significantly just before disappearance under all conditions, and again this increase was not affected by inhibition of myosin V (*Figure 6D*, *Table 1*), suggesting that the activity-induced increase in vesicle mobility was not driven by myosin V.

Finally we examined the changes in vesicle angular displacement immediately prior to disappearance relative to the beginning of observation, which provides a measure of how directed the vesicle motion is (*Gramlich and Klyachko, 2017*). The instantaneous angular displacement was the same under all conditions and was not affected by inhibition of myosin V (*Figure 6E*, *Table 1*).

The combined total displacement, velocity, and angular displacement results suggest that vesicles that are initially tethered at the plasma membrane, become un-tethered and more mobile just before tracking is lost. Activity regulates this process by increasing the vesicle retention (by strengthening tethering) at the membrane pool in a myosin V dependent manner. These analyses further indicate that myosin V does not function in this process to drive vesicle transport away from the membrane pool. Thus the requirement for myosin V in activity-dependent vesicle retention strongly suggests the role for myosin V as a vesicle tether at the plasma membrane.

## Discussion

We took advantage of nanoscale detection of individual vesicle fusion events together with single-vesicle tracking during recycling to examine the mechanisms governing release site refilling in hippocampal synapses. Our results support three key observations: (i) Myosin V plays a major role in refilling of the release sites during repetitive stimulation, but not directly in vesicle release process. Myosin V also regulates spatial distribution of release by preferentially promoting release at more central release sites. (ii) Recycling vesicles undergo a continuous bi-directional shuttling between a membrane pool and an inner pool and the rate of vesicle 'undocking' but not 'docking' is regulated by neural activity (iii) myosin V functions as a tether that retains vesicles at the plasma membrane in

an activity-dependent manner, rather than a motor driving vesicle transport to the release sites. These results uncover a complex dynamic mechanism that governs vesicle availability for release in central synapses.

## Myosin V in release site refilling

As highly processive motors, class V myosins have been recognized to play multiple roles in synaptic development and function, particularly in dendrites (*Kneussel and Wagner, 2013*; *Rudolf et al., 2011*). Myosin Va supports neuronal mRNA transport (*Yoshimura et al., 2006*), while myosin Vb mediates transport of recycling endosomes into dendritic spines and is required for induction of several key forms of long-term synaptic plasticity (*Rudolf et al., 2011*). Although presynaptic terminals are also rich in actin filaments, the role of actin-dependent transport and specifically myosin V in presynaptic processes remains debatable (*Kneussel and Wagner, 2013*). Myosin Va is the only member of the myosin family identified as a synaptic vesicle associated protein and is known to directly interact with the SNARE machinery, including syntaxin 1A and synaptobrevin, in a calcium-dependent manner (*Krementsov et al., 2004*; *Ohyama et al., 2001*; *Prekeris and Terrian, 1997*; *Watanabe et al., 2005*). Yet the role(s) of myosin V/Va in presynaptic processes remains controversial due to contradictory results from different knockout studies (*Kneussel and Wagner, 2013*; *Rudolf et al., 2011*; *Schnell and Nicoll, 2001*; *Yoshii et al., 2013*). This discrepancy could arise from developmental or compensatory effects and from the difficulty of separating pre- and postsynaptic effects of myosin V inhibition in electrophysiological recordings of postsynaptic currents. We were able to bypass both of these complications using direct recordings of vesicle release and recycling in presynaptic boutons together with an acute approach to inhibit myosin V in developed neurons.

Myosin V has been implicated in the exocytosis of secretory granules in non-neuronal cells (*Eichler et al., 2006*; *Porat-Shliom et al., 2013*; *Rudolf et al., 2011*) and of large dense-core vesicles in neurons (*Bittins et al., 2009*; *Kögel et al., 2010*). Here we found that in the case of synaptic vesicles, the process of exocytosis itself was not directly affected by myosin V inhibition in hippocampal synapses. In contrast, our results support a major presynaptic role for myosin V in the release site refilling process as a vesicle tether at the plasma membrane. This function of myosin V is supported by our EM results demonstrating a marked docking defect caused by inhibition of myosin V during sustained stimulation. This is in line with the published EM observations that the number of docked secretory granules in neuroendocrine cells is reduced by myosin V inhibition (*Desnos et al., 2007*). Importantly, our finding of increased retention/tethering of vesicles at the membrane during high-frequency stimulation is consistent with the calcium-dependent interaction of myosin V with the SNARE machinery (*Krementsov et al., 2004*; *Ohyama et al., 2001*; *Prekeris and Terrian, 1997*; *Watanabe et al., 2005*). While elucidating the specific molecular interaction mediating this function of myosin V is beyond the scope of the current study, the submicromolar calcium-dependent binding of myosin V to syntaxin-1A (*Watanabe et al., 2005*) may serve as a mechanism by which vesicles are specifically targeted to the release sites at the plasma membrane.

We note that our results do not argue against a possible additional role of actin cytoskeleton and myosin motors in transporting vesicles during some of the earlier steps of the recycling process, prior to vesicle tethering/docking. Indeed, we and others described a directed actin-dependent component of vesicle motion (*Forte et al., 2017*; *Kisiel et al., 2014*; *Peng et al., 2012*) and another member of the myosin family, myosin II, has been implicated in supporting multiple stages in the vesicle recycling, including vesicle translocation (*Chandrasekar et al., 2014*; *Chandrasekar et al., 2013*; *Hayashida et al., 2015*; *Kisiel et al., 2014*; *Miki et al., 2016*; *Peng et al., 2012*; *Seabrooke et al., 2010*; *Takagishi et al., 2005*). Notably, myosin II does not possess a significant vesicle-transporting ability (*Kneussel and Wagner, 2013*; *Porat-Shliom et al., 2013*) and is likely to act indirectly by generating tension and promoting actin dynamics which is required for processive motion of other myosin isoforms (*Semenova et al., 2008*). In contrast, myosin V is a highly processive motor and we previously found that it supports vesicle transport in the axon during inter-synaptic vesicle exchange in central neurons (*Gramlich and Klyachko, 2017*). Inhibition of myosin V also slightly, but significantly, reduced directionality of vesicle motion inside the synaptic boutons (*Gramlich and Klyachko, 2017*). Myosin V may thus play a dual role in vesicle recycling both as a tether at the release sites and as a transporting motor. Yet the specific stage(s) of the recycling

process mediated by actin-dependent transport and myosin V or other members of the myosin family remains to be elucidated.

## Myosin V in the differential engagement of central vs peripheral release sites

Our analyses of spatial distribution of release events within individual AZs suggest that central release sites are engaged more frequently under basal conditions and this preferential re-use is determined, in part, by myosin V. Moreover, we previously found that during high-frequency stimulation reuse of release sites shifts towards periphery, and here we observed that this shift is exacerbated by inhibition of myosin V. These results suggest that release sites have a spatial gradient of basal reuse probability from the center to periphery, and that the spatial shift in release site reuse is activity- and myosin V-dependent. Our analyses suggest that this spatial shift is unlikely to arise from myosin V involvement in endocytosis. The center vs periphery differences in release site reuse may arise, for example, if density of actin filaments is larger near the center of the AZ, which could increase the basal probability of vesicle tethering at the more central release sites. A shift of release towards periphery during stimulation could arise because central release sites, which are engaged first, would have a reduced availability during the progression of sustained high-frequency activity. It remains to be determined whether the spatial effect of myosin V inhibition is a reflection of myosin V's function in vesicle tethering or relies on an independent mechanism. Notably, differential spatial distribution of functionally distinct modes of release has been observed for kiss-and-run (closer to the AZ center) vs full fusion events (more peripheral) (*Park et al., 2012*). While our tools do not currently permit distinguishing these modes of release, our findings of an activity-dependent shifts toward increased reuse of peripheral release sites is consistent with the shift from prevalent kiss-and-run to full fusion observed with increase in calcium elevation or increased stimulation frequency (*Harata et al., 2006*; *Richards, 2010*; *Zhang et al., 2009*). Our results further predict that by spatially controlling release site reuse, myosin V may play a role in setting the balance between kiss-and-run and full fusion modes of release.

## A myosin V- and activity-dependent reversible vesicle tethering in release site refilling

A widely held view of vesicle recycling in central synapses assumes a unidirectional flow of vesicles from the sites of endocytosis to the RRP directly or via another recycling/reserve pool(s) (*Rizzoli, 2014*). Our results suggest an important revision to this view with a model of sustained bidirectional vesicle shuttling between a membrane pool and an inner pool. Within this model, vesicles that are tethered at the membrane pool by myosin V have a residential time on the order of twenty seconds in resting conditions and undergo repeated cycles of detaching and transitioning to the inner pool and back to the membrane. Our results further indicate that elevated activity shifts the equilibrium towards vesicle retention at the plasma membrane in a Myosin V-dependent manner by reducing the detachment rate 2 – 3 fold, presumably by strengthening myosinV/SNARE interactions. Although to the best of our knowledge this bi-directional shuttling process has not been described in central synapses, vesicle undocking has been reported in hippocampal synapses to occur at rates three times higher than the occurrence of spontaneous exocytosis (Murphy and Stevens, 1999). Taking the rate of spontaneous exocytosis of ~1 – 2 events per minute at 37°C under our experimental conditions (*Peng et al., 2012*), this gives a rough estimate for the vesicle residence times at the plasma membrane of ~10 – 20 s, similar to our results. Our observations are also reminiscent of the vesicle docking/undocking observed in ribbon synapses, which had an escape rate of 1 in 4 s, comparable to our estimates (*Chen et al., 2013*; *Zenisek, 2008*; *Zenisek et al., 2000*). Notably, much faster vesicle 'approach and bounce' events have also been observed in the Calyx of Held synapses using TIRF with a residence time of 88 ms (*Midorikawa and Sakaba, 2015*). Such rapid events, if present in our model system, would not be reliably detectable in our measurements because their duration is comparable to our time resolution. Interestingly, the same single-vesicle tracking measurements in the calyx of Held revealed a tethering step prior to vesicle docking with a time constant of 3 – 4 s (*Midorikawa and Sakaba, 2015*) which is comparable to our estimates for the vesicle residence time at the membrane pool. Thus cycles of vesicle approaching, tethering and reversal might be a common feature in different classes of synapses. The two-pool shuttling observed here could

also provide a mechanistic basis for the two-step actin-dependent model of RRP refilling, which was recently proposed for cerebellar synapses (*Miki et al., 2016*).

The important limitation of the current study is in defining the functional identity of the two vesicle pools involved in vesicle shuttling/tethering. Our EGTA experiments provide evidence that the activity/myosin V-dependent pool is localized in the close proximity to the sources of calcium influx, that is the plasma membrane, but to what extent this pool represents the functionally defined RRP remains to be determined. Our estimate of the relative sizes of the two pools is also comparable to the differences between functional RRP and recycling pools defined previously (*Harata et al., 2001*). Nevertheless, our single vesicle tracking measurements do not distinguish whether vesicles travel along or perpendicular to the AZ plane. The effect of EGTA could thus in principle be also explained by two vesicle pools both localized at the AZ, but at a systematically distinct distance to the source of calcium influx, that is calcium channels. However, this model is unlikely for two reasons: (i) a large proportion (~60%) of the calcium channels are mobile in the AZ plane with the median surface area explored by individual channel molecules of ~200 – 250 nm (*Schneider et al., 2015*) comparable to dimensions of the entire AZ (*Schikorski and Stevens, 1997*); (ii) the large spatial domain of vesicle motion that we observed in our tracking experiments, which commonly spans several hundred nanometers over our observation period and includes epochs of fast diffusion and directed motion (*Forte et al., 2017*); such extensive vesicle mobility is difficult to reconcile with the motion restricted along the AZ given the very limited AZ dimensions and the vesicle crowding at the AZ which unlikely to allow fast diffusion or directed motion. Further studies are needed to define the precise spatial localization of these two pools and their correspondence to the RRP and recycling pool. Revealing spatial identity of these pools would require a 3D localization of the AZ with a fluorescence marker in each individual bouton and a simultaneous dual-color imaging of vesicle motion/tethering. Such measurements are currently difficult to perform because even a very small bleed through of the AZ signal to the vesicle channel increases the background noise and reduces localization accuracy. Furthermore, in the absence of a single labeling method that permits simultaneous monitoring of both vesicle motion and release, it is not currently feasible to unambiguously combine these two measurements for the same vesicle, and thus to directly link vesicle motion and tethering inside the synaptic boutons with the functional vesicle pool identity. Nevertheless, our findings provide the first step towards understanding of the mechanisms governing release site refilling and vesicle availability for release.

## Materials and methods

**Key resources table**

| Reagent type (species) or resource | Designation | Source or reference | Identifiers | Additional information |
|---|---|---|---|---|
| Chemical compound, drug | MyoVin-1 (Myo-1) | EMD Millipore | Cat. No. 475984 | |
| Chemical compound, drug | Pentabrom opseudilin (PBP) | FISHER SCIENTIFIC | Cat. No. 501015859 | |
| Chemical compound, drug | EGTA-AM | FISHER SCIENTIFIC | Cat. No. E1219 | |
| Chemical compound, drug | SGC5 | VWR | Cat. No. 89410–772 | |
| Chemical compound, drug | DL-AP5 Sodium salt | Tocris | Cat. No. 3693 | |
| Chemical compound, drug | CNQX disodium salt | Tocris | Cat. No. 1045 | |

*Continued on next page*

*Continued*

| Reagent type (species) or resource | Designation | Source or reference | Identifiers | Additional information |
|---|---|---|---|---|
| Chemical compound, drug | HEPES | Sigma | Cat. No. H4034 | |
| Chemical compound, drug | D-(+)-Glucose | Sigma | Cat. No. G 7021 | |
| Chemical compound, drug | Calcium chloride dihydrate | Sigma | Cat. No. 223506 | |
| Chemical compound, drug | Magnesium chloride hexahydrate | Sigma | Cat. No. M9272 | |
| Chemical compound, drug | Minimum Essential Media (MEM) - No Phenol Red | thermofisher | Cat. No. 51200–038 | |
| Chemical compound, drug | Characterized Fetal Bovine Serum | hyclone | Cat. No. SH30071.03 | |
| Chemical compound, drug | Penicillin-Streptomycin (5,000 U/mL) | thermofisher | Cat. No. 15070063 | |
| Chemical compound, drug | N-2 Supplement (100X) | thermofisher | Cat. No. 17502048 | |
| Chemical compound, drug | Donor Equine Serum | hyclone | Cat. No. SH30074.03 | |
| Chemical compound, drug | Sodium Pyruvate (100 mM) | thermofisher | Cat. No. 11360–070 | |
| Chemical compound, drug | Neurobasal-A Medium | thermofisher | Cat. No. 10888–022 | |
| Chemical compound, drug | B-27 Supplement (50X), serum free | thermofisher | Cat. No. 17504–044 | |
| Chemical compound, drug | GlutaMAX Supplement | thermofisher | Cat. No. 35050061 | |
| Chemical compound, drug | Earle's Balanced Salts | sigmaaldrich | Cat. No. E3024 | |
| Chemical compound, drug | Corning Collagen I, Rat | Fisher Scientific | Cat. No. 354236 | |
| Chemical compound, drug | Cover Glasses | Fisher Scientific | Cat. No. 12-545-80 | |
| Chemical compound, drug | Papain | Worthington Biochemical | Cat. No. LS003126 | |
| Chemical compound, drug | PDL (poly-D-lysine) | BD Biosciences | Cat. No. 40210 | |
| Chemical compound, drug | Trypsin-EDTA (0.05%), phenol red | thermofisher | Cat. No. 25300–054 | |

*Continued on next page*

*Continued*

| Reagent type (species) or resource | Designation | Source or reference | Identifiers | Additional information |
|---|---|---|---|---|
| Biological sample (*Rattus norvegicus*, Female) | Sprague-Dawley Timed-Pregnant rat, E15 pups | Charles River | | Pups of both genders |
| Software, algorithm | MATLAB | MathWorks | RRID:SCR_001622 | |
| Software, algorithm | u-track2.0 | Jaqaman, K., et al. (2008), Nat.Meth. 5, 695–702 | Gaudenz Danuser Lab | |
| Software, algorithm | ImageJ | https://imagej.nih.gov/ij/ | RRID:SCR_003070 | |
| Software, algorithm | Fiji | http://fiji.sc | RRID:SCR_002285 | |
| Software, algorithm | inkscape | | RRID:SCR_014479 | |
| Other | VGluT1-pHluorin | Drs. Robert Edwards and Susan Voglmaier (UCSF) | | Genetically encoded optical indicator of vesicle release and recycling |

## Neuronal cell cultures

Neuronal cultures were produced from the hippocampus of E16-17 rat pups of mixed gender as previously described (*Peng et al., 2012*). Hippocampi were dissected from E16-17 pups, dissociated by papain digestion, and plated on coated glass coverslips containing an astrocyte monolayer. Neurons were cultured in Neurobasal media supplemented with B27. All animal procedures conformed to the guidelines approved by the Washington University Animal Studies Committee (protocol approval # 20170233).

## Lentiviral infection

VGlut1-pHluorin was generously provided by Drs. Robert Edward and Susan Voglmaier (UCSF) (*Voglmaier et al., 2006*). Lentiviral vectors were generated by the Viral Vectors Core at Washington University. Hippocampal neuronal cultures were infected at DIV3.

## Fluorescence microscopy

### Neurotransmitter release measurements

All experiments were conducted at 37°C within a whole-microscope incubator (In Vivo Scientific) at DIV16–19 as we described previously (*Maschi and Klyachko, 2017*). Neurons were perfused with bath solution (125 mM NaCl, 2.5 mM KCl, 2 mM CaCl2, 1 mM MgCl2, 10 mM HEPES, 15 mM Glucose, 50 mM DL-AP5, 10 mM CNQX, pH adjusted to pH 7.4). Fluorescence was excited with a Lambda XL lamp (Sutter Instrument) through a 100 × 1.45 NA oil-immersion objective and captured with a cooled EMCCD camera (Hamamatsu). With this configuration the effective pixel size was 80 nm. Focal plane was continuously monitored, and focal drift was automatically adjusted with 10 nm accuracy by an automated feedback focus control system (Ludl Electronics). Field stimulation was performed by using a pair of platinum electrodes and controlled by the software via Master-9 stimulus generator (A.M.P.I.). Images were acquired using two frames with an acquisition time of 40 ms, one 45 ms before stimulation and one coincidently (0 ms delay) with stimulation. In some experiments (*Figure 2A-C*) imaging was performed using a cooled sCMOS camera (Hamamatsu). With this configuration, the effective pixel size was 60 nm and images were acquired at a frame rate of 200 ms.

## Single-vesicle tracking

Sparse vesicle labeling and functional synapse localization were performed following our previously developed procedures (*Forte et al., 2017*; *Gramlich and Klyachko, 2017*; *Peng et al., 2012*) The same bath solution as above was used for the dye loading and imaging, except 0.2 mM CaCl$_2$, 1.0 mM MgCl$_2$ were used to wash excess dye from the sample. 10 µM SGC5 (Biotium) were added to the bath solution for the dye loading step. Samples were imaged for 50 – 70 s, at an exposure rate of 80 msec (with a total frame rate of 10 Hz). Samples were stimulated for 10 s at 20 Hz with a 10 s delay after the first frame.

## Pharmacology

MyoVin-1 (Millipore), Pentabromopseudalin (PBP, Fisher Scientific) or EGTA-AM (Millipore) were diluted in DMSO (Sigma-Aldrich) and stored at −20°C. Samples were incubated in imaging solution with 30 µM Myo-1 for 5–10 min or 5 µM PBP for 5 min, or 250 µM EGTA-AM for 20 min before dye loading. The effective final DMSO concentration was <0.5%. Extended exposure to MyoVin-1 or PBP caused cell death, thus the bath solution during the experiment did not include Myo-1 or PBP. Our control measurements indicated that continuous presence of these blockers during the experiments did not have additional effects on vesicle motility beyond the effects of pre-incubation (data not shown).

## Large-Area scanning electron microscopy (LaSEM)

Cultures were fixed in a solution containing 2.5% glutaraldehyde and 2% paraformaldehyde in 0.15 M cacodylate buffer with 2 mM CaCl2, pH 7.4 that had been warmed to 37 °C for one hour. In experiments with KCl-induced depolarization, fixation was performed immediately following KCl application, and care was taken to complete the fixation procedure within a few seconds. The samples were then stained according the methods described by *Deerinck et al., 2010*. In brief, coverslips were rinsed in cacodylate buffer 3 times for 10 min each, and subjected to a secondary fixation for one hour in 2% osmium tetroxide/1.5% potassium ferrocyanide in cacodylate buffer for one hour, rinsed in ultrapure water 3 times for 10 min each, and stained in an aqueous solution of 1% thiocarbohydrazide for one hour. After this, the coverslips were once again stained in aqueous 2% osmium tetroxide for one hour, rinsed in ultrapure water 3 times for 10 min each, and stained overnight in 1% uranyl acetate at 4 °C. The samples were then again washed in ultrapure water 3 times for 10 min each and en bloc stained for 30 min with 20 mM lead aspartate at 60 °C. After staining was complete, coverslips were briefly washed in ultrapure water, dehydrated in a graded acetone series (50%, 70%, 90%, 100% x2) for 10 min in each step, and infiltrated with microwave assistance (Pelco BioWave Pro, Redding, CA) into Durcupan resin. Samples were flat embedded in a polypropylene petri dish and cured in an oven at 60 °C for 48 hr. Post resin curing, the coverslips were exposed with a razor blade and etched off with concentrated hydrofluoric acid. Small pieces of the resin containing the cells was then cut out by saw and mounted onto blank resin stubs before 70 nm thick sections were cut in the cell culture growing plane and placed onto a silicon wafer chips. These chips were then adhered to SEM pins with carbon adhesive tabs and large areas (~330×330 µm) were then imaged at high resolution in a FE-SEM (Zeiss Merlin, Oberkochen, Germany) using the ATLAS (Fibics, Ottawa, Canada) scan engine to tile large regions of interest. High-resolution tiles were captured at 16,384 × 16,384 pixels at 5 nm/pixel with a 5 µs dwell time and line average of 2. The SEM was operated at 8 KeV and 900 pA using the solid-state backscatter detector. Tiles were aligned and export using ATLAS 5.

## Image and data analysis

### Vesicle fusion analyses

#### Fusion event localization

The fusion event localization at subpixel resolution was performed using MATLAB and the uTrack software package that was kindly made available by Dr. Gaudenz Danuser lab (*Aguet et al., 2013*; *Jaqaman et al., 2008*). The input parameters for the PSF were determined using stationary green fluorescent 40 nm beads. See (*Maschi and Klyachko, 2017*) for further details.

## Whole synapse intensity quantification

Individual synapses were identified as fluorescence intensity peaks that increased in intensity upon stimulation using Matlab. Whole-synapse vGlut1-pHluorin intensity was measured over a region of interest (ROI) 10 pixels (600 nm) in radius from the center of each bouton. For a given stimulation frequency, five trains at 20 s intervals were applied. The data was pooled for the five consecutive trains in the same bouton because no difference between subsequent trains was observed (*Figure 2—figure supplement 1F,G,H*) and because the magnitude of changes measured using only the first train was indistinguishable from that averaged across trains (*Figure 2—figure supplement 1G, H*).

## Hierarchical clustering analysis

Hierarchical clustering was performed using built-in functions in MATLAB as we described previously (*Maschi and Klyachko, 2017*).

## Release site re-use probability analysis

a two-pulse stimulation paradigm was used as we described previously (*Maschi and Klyachko, 2017*). For each pair of stimuli, a subset of boutons was identified in which release events were detected for both stimuli in the pair; the probability of observing the two release events at the same release site was then calculated.

## EM analyses

### Synapses identification for EM analysis

we used three characteristic features for manual synapse identification: the presence of a synaptic vesicle cluster, the postsynaptic density and the uniform synaptic gap between pre and postsynaptic membranes.

### AZ to closest vesicle analysis of EM data

membrane opposite to the PSD was divided in small sections (0.5 nm) and the distance from each small section to the closer vesicle was determined using Matlab. This subset of membrane 'closest' vesicles was subsequently used to determine the relation between 'docked' and 'tethered' vesicle fractions. 'Docked' vesicles were defined as those with the distance from the membrane to the vesicle center less than 30 nm and 'tethered' vesicle as those with the distance less than 100 nm (*Figure 3B*).

## Single-Vesicle detection and tracking

### Single-vesicle tracking

The feature identification and subpixel localization were performed using MATLAB and uTrack software package kindly provided by Dr. Gaudenz Danuser lab (*Aguet et al., 2013*; *Jaqaman et al., 2008*) following our previously developed procedures (*Forte et al., 2017*; *Gramlich and Klyachko, 2017*; *Peng et al., 2012*). Localization of functional synapses was performed using ImageJ. Quantification of vesicle motion was performed using the three-frame moving average of vesicle position to mitigate the effects of noise.

### Integrated synapse intensity analysis

We quantified the total integrated intensity of a synapse containing a single labeled vesicle to determine changes in intensity after vesicle detection was lost. Analysis was performed in ImageJ. First, we drew a 10 × 10 pixel box (800 × 800 nm), corresponding to the typical size of a synapse (*Schikorski and Stevens, 1997*, *Schikorski and Stevens, 1999*), around the center of a vesicle at T = 0. Second, we summed the intensity of all pixels within the box for each frame in the movie (I(T)). Third, we normalized the data by the first frame intensity (I(T)/I(0)).

### Vesicle detection with reduced stringency

To determine whether disappearing vesicles could be tracked longer with less stringent detection criteria, we loosened the full width at half max (FWHM) restriction on the width of the vesicle image by a factor of 2 (from 1.5 to 3 pixels). Consequently, we were able to track disappearing vesicles

longer (*Figure 4C* and *Figure 4—figure supplement 1E,F*), and also track few vesicles that left their synapses and traveled to the axon (*Figure 4—figure supplement 1C,D*).

## Identification of vesicles that exited from synapse

We determined if vesicle detection was lost because vesicles exited the synapse to the axon using definitions and analysis we described previously (*Gramlich and Klyachko, 2017*) (*Figure 4—figure supplement 1A,B*). To quantify the change in integrated synapse intensity upon vesicle exit to the axon, we used tracks of the vesicles known to exit a synapse without loss of detection: *Figure 4—figure supplement 1A,B* shows the intensities of 4 different synapses with apparent vesicle exit, aligned relative to vesicle exit, so that T = 0 is defined as the time a vesicle is >600 nm from its position in the first frame (*Gramlich and Klyachko, 2017*). The intensity remains constant before exit followed by a 20–30% reduction after the vesicle exits the synapse. The rate at which the intensity drops depends upon the speed of the vesicle as it exits the synapse, with faster vesicles exhibiting a faster drop in intensity. Thus we used the average curve (black line in *Figure 4—figure supplement 1B*) as the baseline to determine if loss of detection was due to vesicle exit from the synapse to the axon. We found three vesicles which disappearance was consistent with the exit to the axon.

## Vesicle disappearance and appearance analysis

The subgroup of vesicle tracks used in the disappearance and appearance rate analysis was determined as follows:

i. All tracks that started at T = 0, were within 600 nm of a synapse and were tracked longer than 50 frames were chosen for disappearance group ($S_1$).
ii. Any synapse that had more than one track at T = 0 was excluded.
iii. Any track that traveled more than 800 nm from the center of a synapse during observation was excluded.
iv. Tracks that appear in a synapse were chosen for the appearance group ($S_2$) only if another track disappeared in the same synapse at an early time point.
v. Appearing tracks were a minimum of 20 frames in duration.
vi. Appearing tracks were not chosen if they began outside of the synapse.

The rate of disappearance was calculated as the cumulative fraction of vesicles which detection was lost in group $S_1$ as a function of time. The number of lost tracks per time ($N_1(t)$) were calculated. Lost tracking was calculated as the number of tracks reporting 'NaN' during tracking. Some tracks have gaps, or missing frames reported as 'NaN,' which results in variance in $N_1(t)$. The total lost tracks per frame was then divided by the total number of tracks in the group:

$$F_1(t) = \frac{N_1(t)}{|S_1|}$$

The rate of appearance was determined as the cumulative fraction of appearances in group $S_2$ as a function of time. At each frame (t), the total number of new track appearances ($N_2(t)$) were counted and added the total number of appearances per time ($F_2(t)$). If a track appeared more than once following another track in the same synapse, then the new appearance was also counted in the cumulative total. Thus, the appearance rate measures the fraction of appearances as a function of time:

$$F_1(t) = \frac{1}{|S_2|}\sum_{i=0}^{t} N_2(t)$$

Error-bars on disappearance and appearance rates were determined from averages of their best fit residuals. Disappearance and Appearance rates were fit to exponential and linear functions, respectively. The difference from the best fit line and raw data was then determined. The average of those differences were then reported with the value for each condition (*Figure 5 C,E*).

## Vesicle disappearance and appearance oversampling correction

Vesicle disappearance and appearance distributions were sampled at a rate of 10 frames per second. However, the typical disappearance rate was on the order of 1 vesicle per second (1 vesicle per 10 frames) resulting in significant oversampling. Thus, we averaged the oversampled distributions

with a five-frame moving average and plotted every fifth data point. Further, we performed statistical analysis on the averaged data to prevent over-sampling bias of the statistics.

## Statistical analyses

Statistical analyses were performed in Matlab. Statistical significance was determined using two tailed Student's t-test, Kolmogorov-Smirnov (K-S) test, or chi-squared test where appropriate. Data is reported as mean ±SEM or±Residual from fits to data, as indicated in the text.

# Acknowledgments

This work was supported in part by grants to VAK from NINDS (RO1 NS105776) and CIMED center at Washington University. We acknowledge the assistance of Matthew Joens and Dr. James Fitzpatrick at the Washington University Center for Cellular Imaging in EM studies.

# Additional information

## Funding

| Funder | Grant reference number | Author |
| --- | --- | --- |
| CIMED Center at Washington University | | Vitaly A Klyachko |
| National Institute of Neurological Disorders and Stroke | NS105776 | Vitaly A Klyachko |

The funders had no role in study design, data collection and interpretation, or the decision to submit the work for publication.

## Author contributions

Dario Maschi, Software, Formal analysis, Investigation, Methodology; Michael W Gramlich, Software, Investigation, Methodology; Vitaly A Klyachko, Conceptualization, Supervision, Funding acquisition, Methodology, Writing—original draft, Project administration

## Author ORCIDs

Vitaly A Klyachko  http://orcid.org/0000-0003-3449-243X

## Ethics

Animal experimentation: All animal procedures were in compliance with the US National Institutes of Health Guide for the Care and Use of Laboratory Animals. All animal procedures conformed to the guidelines approved by the Washington University Animal Studies Committee (protocol approval # 20170233).

## Decision letter and Author response

Decision letter https://doi.org/10.7554/eLife.39440.016
Author response https://doi.org/10.7554/eLife.39440.017

# Additional files

## Supplementary files

• Transparent reporting form
DOI: https://doi.org/10.7554/eLife.39440.013

## Data availability

All data generated or analyzed during this study are included in the manuscript and supporting information provided.

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

## Appendix 1

DOI: https://doi.org/10.7554/eLife.39440.014

## Computational model of vesicle disappearance

We computationally simulate vesicle disappearance rates following a previously published dynamic Monte-Carlo model (*Gramlich and Klyachko, 2017*), using the python 3.3 language. First, we defined the following variables:

| | |
|---|---|
| $P_{\delta,0}$ | Baseline disappearance probability |
| $P_{\delta,S}$ | Disappearance probability during stimulation |
| $T_0$ | Time of vesicle disappearance |
| $T_S$ | Time stimulation starts |
| $N_{tot}$ | Total number of vesicles simulated |
| $F_{labeled}$ | Fraction of $N_{tot}$ vesicles labeled at $T_0$ |

Second, at the beginning of each time-step, we obtained two random numbers ($R_1$, $R_2$) between [0,1] using the python numpy random number generator random.rand(). Third, if $R_1$ is less than $P_{\delta,0/S}$ and $R_2$ is less than the fraction of current labeled vesicles *then* a labeled vesicle is chosen to be removed and replaced with an un-labeled vesicle; otherwise an un-labeled vesicle is chosen to be replaced with another un-labeled vesicle.

The fraction of labeled vesicles remaining in the simulation was plotted as a function of time-steps. The fraction of simulated vesicles that disappeared was then compared to the experimentally observed fractions in (*Figure 5B*). When best-fit probability was found that matched experimentally observed data, the simulated δ-rate was calculated:

$$\delta = \frac{\text{Probability of vesicle disappearance per frame}}{\text{Number of simulated vesicles}} = \frac{p_\delta}{N_{Total}}$$

## Calculation of the initial fraction of vesicles in Pools I and II.

To calculate the fraction of vesicles in each pool at T = 0 based on the effect of EGTA-AM on the basal disappearance rate the following definitions/assumptions were made:

1. The true disappearance rate (from Pool I) is ($\delta_{CT}$)
2. The observed rate of disappearance in the presence of EGTA-AM is ($\delta_{EGTA}$), the observed rate of appearance is ($\alpha_{EGTA}$).
3. $\rho$ is a fraction of labeled vesicles in Pool I; thus 1- $\rho$ is a fraction of labeled vesicles in Pool II.
4. Observed changes in disappearance rate are due to contribution from vesicles in Pool II.
5. The change in observed rate of disappearance in the presence of EGTA-AM ($\delta_{EGTA}$) is determined by a combination of a vesicle fraction in Pool I ($\rho$) and a contribution from the vesicle fraction in Pool II (1- $\rho$).

These assumptions can be combined in an equation:

$$\delta_{EGTA} = \rho^* \delta_{CT} + (1-\rho)^* \alpha_{EGTA}$$

This equation results in the solution:

$$\rho = \frac{\delta_{EGTA} - \alpha_{EGTA}}{\delta_{CT} - \alpha_{EGTA}}$$

Using the range of values observed experimentally, we obtained:
$\delta_{EGTA}$ = 0.038 ± 0.006 fraction per second;

$\alpha_{EGTA}$ = 0.014 ± 0.005 fraction per second;
$\delta_{CT}$ = 0.047 ± 0.017 fraction per second;
$\rho$ = 74.5 ± 11%

