## [Decision Letter]

Thank you for submitting your article "Myosin V functions as a vesicle tether at the plasma membrane to control neurotransmitter release in central synapses" for consideration by *eLife*. Your article has been reviewed by three peer reviewers, one of whom is a member of our Board of Reviewing Editors, and the evaluation has been overseen by Richard Aldrich as the Senior Editor. The following individual involved in review of your submission has agreed to reveal his identity: Alain Marty (Reviewer #3).

The reviewers have discussed the reviews with one another and the Reviewing Editor has drafted this decision to help you prepare a revised submission.

Summary:

Here Maschi et al. examine how active zone release sites are replenished with new vesicles following exocytosis, specifically focusing on the role for myosin V in this process, using nano-scale imaging to track the behavior of single vesicles. They identify that vesicle replenishment is sensitive to myosin V inhibition and provide evidence that vesicle undocking from the plasma membrane – i.e. the rate of vesicle exit from the field of view – is suppressed by activity, which in turn, helps strengthen the tether in a myosin V-dependent manner. The study is timely and imaginative, and it reports a series of provocative new observations. Methods are appropriate, analysis and modeling are generally well done (except for some caveats below), and the writing is clear. Overall, the study should be of general interest to the readership of *eLife*. However, in its present form, the paper has two major issues.

Firstly, myosin V inhibitor is shown to have effects on the measurements of single release site exocytosis (Figure 1), activity-dependent vesicle recruitment to the release site (Figure 2), and single vesicle tracking inside the presynaptic terminal (Figure 3-4). The authors present the case that these effects are related to each other. However, there remains a possibility that such effects represent distinct actions of myosin V that are based on interactions with different molecular processes, similarly to a role for myosin V in inter-bouton trafficking of vesicles that have been previously reported by the Klyachko group. For example, the results in Figure 1-2 suggest a very local action of myosin whereas the results of Figure 3-4 suggest a longer range action. Secondly, the analysis of vesicular movement based on standard deviation, and the accompanying double pool model contains several weakness that require careful consideration. The specific points relating to the concerns are outlined below. The reviewers have agreed that these concerns are addressable, in principle, by additional experiments, re-analysis of data, and re-writing of the text over a two-month period.

Essential revisions:

1) Figure 2: The two pool model involving tethered vesicles and vesicles remote from the docked pool (inner recycling pool or reserve pool) is not necessarily compatible with myosin V inhibitor-dependent spatial distribution of release site re-use (Figure 2D-F), and this point requires further clarifications. What is the relationship between myosin V-dependent tethering and the location of release sites with respect to VGCCs responsible for the calcium influx that drives exocytosis? An attractive interpretation of the results in Figure 2D-E would be that myosin V is involved in endocytosis. By inhibiting endocytosis, Myo1 would then enhance the size of the AZ in an activity dependent manner, thus increasing the distance of peripheral release sites to the center.

2) Figure 2A-C: A control should be added showing the (lack of?) effects of a previous stimulation in control conditions, without addition of Myo1.

3) Figure 2H-I: The control with Myo1 but no KCl is missing.

4) Figure 3 and 4: Orientation of the synapse with respect to the imaging plane. The analysis makes the implicit assumption that the AZ is orthogonal to the Z-axis. In Figure 1, the argument can be made that AZs displaying several release sites must be mainly orthogonal to the Z-axis, introducing a sample bias in favor of synapses displaying a horizontal orientation. With the protocol of Figure 3-4 no such argument can be made, so that a change in the Z-axis may represent a movement parallel to the AZ plane. This challenges the identification of pool I and pool II as representing two sets of SVs positioned at different distances of the AZ membrane.

In addition to the assumption of a horizontal orientation, the analysis implicitly assumes that the imaged SV is initially in pool I, near the AZ membrane. Since pool II is larger than pool I, it would seem more likely that it would be initially in pool II, away from the AZ membrane. One way out of this conundrum may be to assume an experimental bias towards pool I by selection of slowly moving vesicles.

5) Figure 3 and 4: Lack of calibration of σ changes. The main argument linking disappearance to z movement is that σ increases. However the extent of z movement associated with the observed change in σ remains unclear. In view of the results of Figure 2 the z movement is expected to take place in a range of 10s of nm, but since the size of the point spread function is on the order of 1 micron in the Z-axis, the z movements corresponding to the observed σ changes are likely to be larger, in a range of 100s of nm. This issue could perhaps be settled by measuring σ values as a function of z for a fluorescent bead of small size.

6) Figure 3 and 4: Interpretation of the σ change. The proposed change in z position is one possibility, but an alternative interpretation is that the SGC5 fluorescence bleaches more rapidly than the background fluorescence. To distinguish between these possibilities, it would be good to see whether σ returns near its initial value when the SV reenters pool I (e.g. in Figure 3C).

7) Figure 3: The data in Figure 3B suggest a continuous drift. By contrast, the scheme of Figure 3D suggests abrupt changes with probabilities linked to α and δ. A possible explanation for the drift could be a movement of the preparation with respect to the imaging plane. Yet another interpretation would be a diffusion process, without the intervention of distinct vesicular pools. This point needs to be clarified.

8) Figure 4: Regarding EGTA's effect on disappearance and appearance rates, in Figure 4B it seems like EGTA has a role in baseline disappearance rate (although it appears to be non-significant, the extent of reduction comparing +Ctrl and +EGTA in Figure 4B-Basal looks similar to the level of change shown in Figure 4D-Activity for the same groups, the significance seems to be a consequence of reduced dispersion in 4D compared to 4B). Would this mean that calcium regulates the stability/immobility of docked synaptic vesicles in basal conditions?

9) The plots showing spatial displacement of vesicles in Figure 4E are very straightforward and allow a fast understanding for the reader. The authors should include the same plot for inhibition of Myosin V (the data is only presented in Table 1), to emphasize the result and make the conclusion easier to understand.

10) Although not mentioned in the present study, the undocking of vesicles was previously characterized by Murthy and Stevens (Murthy and Stevens, 1999) in hippocampal synapses, who reported that undocking was three times more prevalent than the occurrence of spontaneous exocytosis, similarly to the present findings.

11) Figure 1: The method used to obtain the control data in the upper panel of Figure 1B should be explicitly stated. Likewise, the method used to generate the plot in Figure 1C should be explained.

12) Figure 2: Here the main issue is time. The reloading mechanism discussed in Figure 1 and Figure 2A-F operates on a time scale of a few seconds. In Figure 2H-I however, a prolonged KCl stimulation (10 min) is applied. Notably, this might be harsh for the cultured neurons, and the authors should state the concentration of KCl used. Furthermore, the time between the end of stimulation and fixation is not specified, and this time is presumably again on the order of minutes. One day the experiments of Figure 2H-I will need to be repeated with the better time resolution of the flash and freeze technique. This is not needed for this paper, but the protocols used for EM experiments should be more precisely outlined, and caution should be applied when discussing the links between EM and single vesicle exocytosis results.

---

## [Author Response]

Essential revisions:1) Figure 2: The two pool model involving tethered vesicles and vesicles remote from the docked pool (inner recycling pool or reserve pool) is not necessarily compatible with myosin V inhibitor-dependent spatial distribution of release site re-use (Figure 2D-F), and this point requires further clarifications. What is the relationship between myosin V-dependent tethering and the location of release sites with respect to VGCCs responsible for the calcium influx that drives exocytosis? An attractive interpretation of the results in Figure 2D-E would be that myosin V is involved in endocytosis. By inhibiting endocytosis, Myo1 would then enhance the size of the AZ in an activity dependent manner, thus increasing the distance of peripheral release sites to the center.

We thank the reviewers for this important point and completely agree. We performed a series of analyses to examine a possibility that Myosin V is involved in endocytosis:

1) We directly examined the reviewers’ suggestion what inhibition of myosin V could increase the size of the AZ in an activity-dependent manner thus increasing the distance of peripheral release sites to the center. Our scanning EM analysis showed that AZ size remained unchanged by myosin V inhibition both at baseline (Figure 1E) and during KCl-induced depolarization(Figure 2—figure supplement 1B). This result suggests that inhibition of myosin V does not cause activity-dependent changes in AZ size.

2) We examined if inhibition of myosin V affects dynamics of endocytosis in live and active synapses. The decay of vGlut1-pHluorin signal following a stimulus train is determined by endocytosis and subsequent vesicle reacidification. Changes in the decay of the vGlut1-pHluorin signal in our measurements can thus be interpreted to reflect predominantly changes in endocytosis (Atluri and Ryan, 2006). We found that inhibition of myosin V had no measurable effect on the decay of the vGlut1-pHluorin signal following 50Hz trains (Figure 2—figure supplement 1C-E).

3) If inhibition of myosin V blocks endocytosis, application of sequential stimulus trains would be expected to cause surface accumulation of VGlut1-pHluorin and a corresponding increase in the bouton fluorescence. We compared the amplitudes of VGlut1-pHluorin signal in five consecutive 50Hz trains separated by 20sec each, and did not observe any measurable changes from one train to the next under control conditions or in the presence of myosin V inhibitors (Figure 2—figure supplement 1F, G, H). While an indirect evidence, this result supports the above findings that acute inhibition of myosin V does not significantly affect vesicle endocytosis under our experimental conditions.

Combined, these results indicate that the shift in spatial localization of release events upon myosin V inhibition is not caused by an increase in the AZ size and is unlikely to be mediated by altered endocytosis. These new findings are now presented in the second paragraph of the Results subsection “Myosin V controls release site refilling during high-frequency stimulation”.

We also note that the spatial shift in the location of release Events upon myosin V inhibition was observed not only during stimulus trains but also at low frequency stimulation (Figure 1F). Under these conditions, release is very infrequent and no changes in the size of the AZ are observed (Figure 1E). The observed shift in the distribution of release events upon myosin V inhibition could thus arise without changes in the physical localization of release sites. This effect could arise, for example, from a reduced release capacity of individual release sites we observed, leading to a more spatially distributed release. Indeed, we previously described a comparable phenomenon of reduced release site re-use capacity and a shift in the distance of release events towards AZ periphery during high-frequency stimulation, and this effect was observed without myosin V inhibition (Maschi and Klyachko, 2017). While we cannot exclude the possibility that the spatial and temporal effects of myosin V inhibition arise from distinct myosin V functions, the above considerations and the parallels with the effects of activity point to a common myosin V function as the simplest and the most logical interpretation of our findings. We have expanded the Discussion of the revised manuscript to include these considerations as follows:

“Our analyses suggest that this spatial shift is unlikely to arise from myosin V involvement in endocytosis. […] It remains to be determined whether the spatial effect of myosin V inhibition is a reflection of myosin V’s function in vesicle tethering or relies on an independent mechanism”.

2) Figure 2A-C: A control should be added showing the (lack of?) effects of a previous stimulation in control conditions, without addition of Myo1.

We thank the reviewers for this point and have performed the suggested analysis in two different and complementary ways:

1) We examined responses to 5 consecutive trains of 20 stimuli at 50Hz used in Figure 2A-B and calculated changes in synaptic responses from one train to the next. We found that the difference between subsequent responses in control conditions was negligible (~2%) (Figure 2—figure supplement 1F). Moreover, the change in response from one train to the next was also indistinguishable in control conditions vs. in the presence of myosin V inhibitors (Figure 2—figure supplement 1F).

2) For each of the 5 consecutive stimulus trains, we plotted the magnitude of changes in synaptic response in the presence of myosin V inhibitors relative to control, and did not observe any measurable differences from one train to another (Figure 2—figure supplement 1G, H).

Finally, we note that the magnitude of changes measured using only the first train was indistinguishable from that averaged across trains (Figure 2—figure supplement 1G, H).

Together these analyses provide strong evidence for lack of measurable effects of previous stimulation in our measurements.

We also realized that the measurements in Figure 2C were not described clearly. Measurements for each frequency were performed in a separate set of dishes and normalized to its own control. Therefore there is no effect of previous stimulation in these experiments carried from one frequency to another. The n values reported in the Table 1 represent the total number of dishes for all frequencies combined, since this entire dataset was used to perform the linear regression analysis shown in Figure 2C. We have clarified the description of this experiment in the Results and the Figure 2 legend.

3) Figure 2H-I: The control with Myo1 but no KCl is missing.

We thank the reviewers for this point and note that this control was actually performed (Figure 3—figure supplement 1A, B), but it was not described clearly. We have improved the description of this experiment in Results subsection “Inhibition of myosin V causes a vesicle docking defect during sustained activity”, to clarify this point.

4) Figure 3 and 4: Orientation of the synapse with respect to the imaging plane. The analysis makes the implicit assumption that the AZ is orthogonal to the Z-axis. In Figure 1, the argument can be made that AZs displaying several release sites must be mainly orthogonal to the Z-axis, introducing a sample bias in favor of synapses displaying a horizontal orientation. With the protocol of Figure 3-4 no such argument can be made, so that a change in the Z-axis may represent a movement parallel to the AZ plane. This challenges the identification of pool I and pool II as representing two sets of SVs positioned at different distances of the AZ membrane.

We thank the reviewers for highlighting this limitation of our model. Given the previous literature on calcium channel distribution and mobility within the AZ (Schneider et al., 2015), we believe the simplest model that accounts for our observations is two pools at different distances from the AZ, for the following reasons:

1) The different exchange rates between the two pools suggest that they are different in size and dynamics.

2) Observation that disappearance and appearance rates are differentially affected by EGTA suggest that the two pools are at different distances from the AZ’s calcium channels.

3) Differential activity-dependence of the exchange rates between the two pools supports the notion that these pools have a different distance to the source of calcium influx, the AZ.

We agree with the reviewers that our vesicle tracking measurements do not distinguish whether vesicles travel along or perpendicular to the AZ plane. The effect of EGTA could thus in principle be also explained by two vesicle pools both localized at the AZ, but at a systematically distinct distance to the source of calcium influx, i.e. calcium channels. However, this model is unlikely for two reasons: (i) a large proportion (~60%) of the calcium channels are mobile in the AZ plane with the median surface area explored by individual channel molecules of ~200-250 nm (Schneider et al., 2015) comparable to dimensions of the entire AZ (Schikorski and Stevens, 1997); (ii) the large spatial domain of vesicle motion that we observed in our tracking experiments, which commonly spans several hundred nanometers over our observation period and includes epochs of fast diffusion and directed motion (Forte et al., 2017); such extensive vesicle mobility is difficult to reconcile with the motion restricted along the AZ given the very limited AZ dimensions and the vesicle crowding at the AZ which unlikely to allow fast diffusion or directed motion.

To address the reviewers’ point we now discuss this limitation of our model and these considerations in the last paragraph of the Discussion subsection “A myosin V- and activity-dependent reversible vesicle tethering in release site refilling”.

In addition to the assumption of a horizontal orientation, the analysis implicitly assumes that the imaged SV is initially in pool I, near the AZ membrane. Since pool II is larger than pool I, it would seem more likely that it would be initially in pool II, away from the AZ membrane. One way out of this conundrum may be to assume an experimental bias towards pool I by selection of slowly moving vesicles.

We thank the reviewers for this point and agree. Because disappearance is the only criteria for the vesicle inclusion in our initial analysis and due to the arbitrary relationship between the focal plane and the AZ at any given synapse, there is no a priori information that would allow us to make the initial assignment of the vesicles to Pool I or Pool II. As a result, our measurements of the disappearance rate reflect a mixture of vesicles that are initially in Pool I and Pool II at the beginning of observation. We considered the reviewers’ suggestion to assume bias towards Pool I by selection of slowly moving vesicles to address this issue. However, we found that it was not possible to group vesicles based on their speeds in any systematic manner because vesicle speeds are highly variable during the observation time due to multiple intertwining epochs of diffusion, directed motion and pausing with speeds of different epochs varying several fold along the individual vesicle trajectories (Forte et al., 2017).

However, our finding that EGTA strongly affects only the vesicle appearance rate (and thus vesicles in Pool II) together with the finding that EGTA also has a small effect on the baseline disappearance rate (which we interpreted within our model as a contribution from vesicles in Pool II) provides a simple way to estimate the ratio of vesicles that start in each pool in beginning of our observations. To perform this analysis, we assumed that only the vesicle appearance rate is truly affected by EGTA (based on our observation that disappearance rate for EGTA and Ctrl during stimulation are the same), while the small effect of EGTA on the baseline disappearance rate (vesicles in Pool I) is caused by contribution from Pool II vesicles. Within these assumptions, the model (Appendix) and EGTA results suggest that among all vesicles assumed to be in Pool I in the beginning of observations, 75 ± 11% actually start in Pool I, while 25 ± 11% are physically located in Pool II. This result is consistent with the Pool II being larger, and having a slower exit rate, because the probability that a given labeled single vesicle in our experiment will leave the Pool II is smaller than if it would be in Pool I and thus less likely to be included in our disappearance analysis. These considerations support our model by demonstrating that the majority of the disappearing vesicles are initially located in Pool I. We have added these considerations to the third paragraph of the Results subsection “Vesicle shuttling represents transitions between a membrane pool and an internal pool”.

5) Figure 3 and 4: Lack of calibration of σ changes. The main argument linking disappearance to z movement is that σ increases. However the extent of z movement associated with the observed change in σ remains unclear. In view of the results of Figure 2 the z movement is expected to take place in a range of 10s of nm, but since the size of the point spread function is on the order of 1 micron in the Z-axis, the z movements corresponding to the observed σ changes are likely to be larger, in a range of 100s of nm. This issue could perhaps be settled by measuring σ values as a function of z for a fluorescent bead of small size.

We thank the reviewers for suggesting the σ calibration experiments and have performed these measurements (Figure 4—figure supplement 1G). This calibration allows us to correlate the observed range of vesicle σ values (~120-150 nm) with the range of detectable vesicle motion along the Z-axis of about 100 nm (Figure 4—figure supplement 1G). However, we also note that this concern likely arose from lack of clarity in our description of the results in Figure 2, rather than a problem with the results themselves. The reviewers referred to the results in Figure 2 to estimate relevant distances for the expected vesicle Z-movements between the two pools. However, these measurements in Figure 2 represent changes in the vesicle distance to the AZ upon myosin V inhibition, but do not represent distances in vesicle localization in Pools I and II under normal conditions. Therefore our data in Figure 2 does not provide any estimates for how far apart the pools are expected to be. We also note that only the σ changes in XY plane are used as one of the detection criteria, with the typical σ value of ~120nm. Most importantly, the actual numerical changes in σ values are not used in any of the analyses in the manuscript. We have clarified these points in the fifth paragraph of the Results subsection “Tracking individual synaptic vesicles during recycling supports a model of continuous 238 vesicle shuttling between two vesicle pool”.

6) Figure 3 and 4: Interpretation of the σ change. The proposed change in z position is one possibility, but an alternative interpretation is that the SGC5 fluorescence bleaches more rapidly than the background fluorescence. To distinguish between these possibilities, it would be good to see whether σ returns near its initial value when the SV reenters pool I (e.g. in Figure 3C).

We thank the reviewers for this point and have performed both of the suggested analyses.

First, we performed the analysis of bleaching and showed that the σ changes were not caused by differential bleaching of vesicle signal vs. background (Figure 4—figure supplement 1I).

Second, we compared the σ changes just before vesicle disappearance and right after reappearance (Figure 4—figure supplement 1H). σ values increased just before disappearance, and then decreased just after re-appearance.

Together these results support our interpretation that vesicle going out of focus due to changes in zposition is the predominant cause of vesicle disappearance/loss of detection.

7) Figure 3: The data in Figure 3B suggest a continuous drift. By contrast, the scheme of Figure 3D suggests abrupt changes with probabilities linked to α and δ. A possible explanation for the drift could be a movement of the preparation with respect to the imaging plane. Yet another interpretation would be a diffusion process, without the intervention of distinct vesicular pools. This point needs to be clarified.

We appreciate the reviewers’ point and we believe it is based on a misnomer we used in the figure description: Y-axis in the original Figure 3B was labeled ∆X, but this represents the localization error of the vesicle position (which can increase with vesicle moving away from the focal plane) and not the actual position itself and therefore does not show drift. We performed extensive controls for the drift in our imaging system and found that the drift of the preparation in our measurements is extremely small (6-8nm) (Maschi and Klyachko, 2017; Forte et al., 2017) and is below the localization uncertainty (~15-20nm). We apologize for the confusion in the labeling of original Figure 3B and have re-labeled the figure axis (the new Figure 4B and 4E) as “localization error” to avoid confusion.

8) Figure 4: Regarding EGTA's effect on disappearance and appearance rates, in Figure 4B it seems like EGTA has a role in baseline disappearance rate (although it appears to be non-significant, the extent of reduction comparing +Ctrl and +EGTA in Figure 4B-Basal looks similar to the level of change shown in Figure 4D-Activity for the same groups, the significance seems to be a consequence of reduced dispersion in 4D compared to 4B). Would this mean that calcium regulates the stability/immobility of docked synaptic vesicles in basal conditions?

We appreciate the reviewers’ point. Our measurements of the disappearance rate reflect a mixture of vesicles that are in Pool I and Pool II at the beginning of observation. Because EGTA only strongly affects the appearance rate (and thus vesicles in Pool II), we interpreted the effect of EGTA on baseline disappearance rate to reflect a mixture of vesicles that are in Pool I and Pool II at the beginning of observation. We estimated the ratio of vesicles that start in each pool (as we described in response to point 4 above), assuming that only the vesicle appearance rate is truly affected by EGTA (based on our observation that disappearance rate for EGTA and Ctrl during stimulation are the same), while the small effect of EGTA on the disappearance rate (vesicles in Pool I) is caused by contribution from Pool II vesicles. Within these assumptions, our results suggest that among all vesicles assumed to be in Pool I in the beginning of observations, 75 ± 11% actually start in Pool I, while 25 ± 11% are physically located in Pool II. As we pointed out in our response to the concern #4 above, this result is consistent with the Pool II being larger, and having a slower exit rate, because the probability that a given labeled single vesicle in our experiment will leave the Pool II is smaller than if it would be in Pool I and thus less likely to be included in our disappearance analysis. We have added these considerations to the third paragraph of the Results subsection “Vesicle shuttling represents transitions between a membrane pool and an internal pool” and Appendix.

9) The plots showing spatial displacement of vesicles in Figure 4E are very straightforward and allow a fast understanding for the reader. The authors should include the same plot for inhibition of Myosin V (the data is only presented in Table 1), to emphasize the result and make the conclusion easier to understand.

We thank the reviewers for this suggestion and have included the same plot for inhibition of myosin V accordingly (Figure 6B, C).

10) Although not mentioned in the present study, the undocking of vesicles was previously characterized by Murthy and Stevens (Murthy and Stevens, 1999) in hippocampal synapses, who reported that undocking was three times more prevalent than the occurrence of spontaneous exocytosis, similarly to the present findings.

We thank the reviewers for bringing this oversight to our attention. We have added the citation and corresponding findings to the first paragraph of the Discussion subsection “A myosin V- and activity-dependent reversible vesicle tethering in release site refilling”.

11) Figure 1: The method used to obtain the control data in the upper panel of Figure 1B should be explicitly stated. Likewise, the method used to generate the plot in Figure 1C should be explained.

We thank the reviewers for this suggestion and have expanded the description of the methods in the Results and in the Materials and methods subsections “Whole synapse intensity quantification” and “Release site re-use probability analysis” accordingly.

12) Figure 2: Here the main issue is time. The reloading mechanism discussed in Figure 1 and Figure 2A-F operates on a time scale of a few seconds. In Figure 2H-I however, a prolonged KCl stimulation (10 min) is applied. Notably, this might be harsh for the cultured neurons, and the authors should state the concentration of KCl used. Furthermore, the time between the end of stimulation and fixation is not specified, and this time is presumably again on the order of minutes. One day the experiments of Figure 2H-I will need to be repeated with the better time resolution of the flash and freeze technique. This is not needed for this paper, but the protocols used for EM experiments should be more precisely outlined, and caution should be applied when discussing the links between EM and single vesicle exocytosis results.

We appreciate the reviewers’ point and have expanded and clarify the protocols for the EM experiments and stated the concentration of KCl (55 mM). We also clarified that the fixation was performed immediately following KCl application with only a few second delay. Finally, we avoided making strong conclusions based on the links between the EM and single-vesicle imaging data.